# Proton-driven alternating access in a spinster lipid transporter

Reza Dastvan [1] ✉, Ali Rasouli[2,3], Sepehr Dehghani-Ghahnaviyeh [2,3], Samantha Gies [1] & Emad Tajkhorshid [2] ✉

Spinster (Spns) lipid transporters are critical for transporting sphingosine-1-phosphate (S1P) across cellular membranes. In humans, Spns2 functions as the main S1P transporter in endothelial cells, making it a potential drug target for modulating S1P signaling. Here, we employed an integrated approach in lipid membranes to identify unknown conformational states of a bacterial Spns from *Hyphomonas neptunium* (*Hn*Spns) and to define its proton- and substrate-coupled conformational dynamics. Our systematic study reveals conserved residues critical for protonation steps and their regulation, and how sequential protonation of these proton switches coordinates the conformational transitions in the context of a noncanonical ligand-dependent alternating access. A conserved periplasmic salt bridge (Asp60[TM2]:Arg289[TM7]) keeps the transporter in a closed conformation, while proton-dependent conformational dynamics are significantly enhanced on the periplasmic side, providing a pathway for ligand exchange.

The bioactive lipid sphingosine-1-phosphate (S1P) plays a key role in regulating the growth, survival, and migration of mammalian cells. S1P is critical for lymphocyte trafficking, immune responses, vascular and embryonic development, metastatic cancer, and bone homeostasis[1–4]. It is produced intracellularly by sphingosine kinases and then released extracellularly to engage in its (patho)physiological roles. Its accumulation is linked to the development/progression of cancer and various other diseases, including atherosclerosis, diabetes, and inflammatory disorders. The Spinster (Spns) lipid transporters of the major facilitator superfamily (MFS) are critical for exporting S1P across cellular membranes[5–7]. The Spns family includes three S1P/sphingolipid transporters (Spns1–3) in humans. Spns1 is associated with lysosomal membranes and is one of the major efflux transporters removing degradation products, including sphingolipids, from the lysosomal lumen to maintain normal size and function. Spns1 also participates in the late stages of autophagy, and its loss of function is associated with human lysosomal storage diseases and neurodegenerative diseases[6]. Though less studied, high Spns3 expression is associated with poor prognosis in chemotherapy patients with acute myeloid leukemia and in renal cancer[8]. Spns2 functions as the main S1P transporter in the plasma membrane of endothelial cells[6,7]. Once released, S1P activates its receptors (S1PR1–5), thus regulating many important cellular functions[1]. Its secretion by Spns2 controls S1P levels in circulatory fluids, boosting immune cell function in lymph, suggesting that Spns2 might be a potential immunosuppressant drug target[9–13]. Spns2 transports the S1P analog FTY720-P, an immunomodulating agent, out of cells to bind the S1P receptors and block lymphocyte egress into the circulation[5]. Interestingly, endothelial cell-specific knockout of Spns2 in mice results in impaired lymphocyte egress and prevents tumor metastasis[1,14]. This strongly suggests that Spns2 could be an effective target for reducing metastases through increasing the efficacy of immunotherapy[1]. Thus, a detailed understanding of the functional mechanism of Spns2 and its regulation are of high significance, as they provide a molecular framework for developing new therapeutic strategies for diseases associated with S1P signaling.

Spns proteins are MFS secondary active transporters that use the electrochemical gradient of proton or sodium ions to drive substrate

[1]Department of Biochemistry and Molecular Biology, Saint Louis University School of Medicine, St. Louis, MO 63104, USA. [2]Theoretical and Computational Biophysics Group, NIH Center for Macromolecular Modeling and Bioinformatics, Beckman Institute for Advanced Science and Technology, Department of Biochemistry, and Center for Biophysics and Quantitative Biology, University of Illinois at Urbana-Champaign, Urbana, IL 61801, USA. [3]These authors contributed equally: Ali Rasouli, Sepehr Dehghani-Ghahnaviyeh. ✉e-mail: reza.dastvan@health.slu.edu; emad@illinois.edu

transport[15]. Crystal structures of several MFS transporters reveal that they possess a 12-transmembrane (TM) helix core consisting of two pseudo-symmetrical domains, namely the N-terminal domain (NTD: TMs 1–6) and the C-terminal domain (CTD: TMs 7–12) (Fig. 1a). A central cavity is formed between the two domains and serves as both the binding site and a translocation pathway for the substrate. MFS transporters usually consist of four pseudo-symmetrical, three-helical repeats[16,17]. "Cavity-helices", the first in each repeat (TMs 1, 4, 7, and 10; Fig. 1), form the cavity. "Gating-helices", the second in each repeat (TMs 2, 5, 8, and 11), are usually long and curved (banana-shaped). Together, these four helices form the side walls of the central cavity and provide a lateral gating mechanism to the membrane. The third group of shorter "support helices" (TMs 3, 6, 9, and 12) do not directly contribute to the cavity but contact the lipid bilayer[15,18]. An alternating access mechanism for substrate transport involves a series of conformational changes that switch access of the ligand binding site between the extracellular and intracellular solutions/membrane leaflets[19]. A transporter often undergoes structural transitions between three major functional states: the inward-facing (IF), the outward-facing (OF), and an intermediate, occluded (O) conformation. In MFS proteins, these states may be stabilized by the formation and breaking of inter- and intra-domain salt bridges[20]. In Spns proteins, the gating helices are kinked and, particularly on their periplasmic/extracellular halves, are inherently flexible due to the large abundance of highly conserved helix-breaking residues like glycine (Fig. 1i–k). The resulting deviation of these "rocker" helices from rigid-body movement in the context of a rocker-switch model of the conformational change between the IF and OF states hints at a noncanonical alternating access mechanism for Spns proteins likely involving a stable O state[21–23].

While an AlphaFold[24] model for the human Spns2 has been generated (Fig. 1c), its experimentally resolved structures have yet to emerge. Structures of a bacterial Spns transporter from *Hyphomonas neptunium* (*Hn*Spns) with high sequence similarity to human Spns2 (~20% sequence identity) have been solved (Fig. 1)[25]. The architecture of the *Hn*Spns core follows the typical MFS fold (Fig. 1a, b). The IF crystal structure of *Hn*Spns (Fig. 1) reveals the binding location of an unspecified substrate in the cavity near a conserved residue Arg42$^{TMI}$, which is locked in position by interactions with two other conserved residues, Glu129$^{TM4}$ and Arg122$^{TM4}$ (Fig. 1b, red circle)[25]. Additionally, the structure suggests a potential proton binding site in the NTD (Fig. 1b, blue circle), comprised of highly conserved residues Asp41$^{TMI}$ (Fig. 1d), Trp102$^{TM3}$ and Thr106$^{TM3}$ (Fig. 1h). Compared with other MFS transporters, the binding cavity in *Hn*Spns has fewer charges (e.g., Arg42$^{TMI}$) and is less hydrophilic, consistent with lipidic or lipophilic nature for the transport substrate, e.g., S1P. More than half of the conserved residues in Spns proteins are located around the inner cavity in *Hn*Spns, with the majority in the NTD (Fig. 1)[25]. Hence, it is predicted that human Spns proteins also transport negatively charged lipophilic small molecules. The *Hn*Spns structure indicates the conserved residue Glu129$^{TM4}$ might serve as the main protonation switch coupling the proton motive force (PMF) to large-scale conformational transitions[25].

The most conserved sequence motif in MFS transporters, motif A (Fig. 1h), is located in an intracellular loop connecting TM2 and TM3 (Fig. 1b)[15,26]. Motif A and its surrounding residues from TM4 and TM11 form a conserved "structural motif A", stabilizing the O or OF conformation[15]. A charge-relay network associated with structural motif A, Asp87$^{motif-A}$-Arg91$^{motif-A}$-Asp142$^{TM4}$, is proposed to regulate the inter-domain charge-dipole interaction between TMs 2 and 11 (Fig. 1a). In addition, a salt bridge might be formed between symmetry-related TMs 5 and 8. The highly conserved Arg122$^{TM4}$ is the signature residue of the B-like motif "RxxxG" in Spns proteins (Fig. 1b, f), although in *Hn*Spns the conserved Gly of the motif is replaced by an alanine[15]. This motif also exists in proton-coupled sugar and sugar acid symporters such as *E. coli* XylE and DgoT[27,28]. The arginine of this motif carries a

buried positive charge, and it is in proximity to all the putative protonation sites inside the central cavity (i.e., Asp41$^{TMI}$ and Glu129$^{TM4}$; Fig. 1b). It is proposed that increasing the strength of the positive electrostatic potential inside the central cavity in the IF state is a common mechanism for promoting the cytosolic proton release step in MFS transporters[15].

Progress in delineating the Spns transport mechanism has been hindered since all the *Hn*Spns structures were IF[25]. Thus, the alternating access mechanism and the proton-dependent conformational dynamics of the transporter are undefined. Here, we employed double electron–electron resonance (DEER; also called PELDOR) spectroscopy[29–33], combined with nonequilibrium, driven[34–39] and DEER-guided molecular dynamics (MD) simulations[40] to identify new conformational states of *Hn*Spns in the membrane and to characterize its proton- and substrate-coupled conformational changes in a lipid bilayer-like environment. Our systematic study reveals the conformational changes of *Hn*Spns following protonation of critical acidic residues, conserved in most Spns proteins and their bacterial homologs (Fig. 1d, f), in the context of a noncanonical ligand-dependent alternating access mechanism.

## Results

Mechanistic characterization of the transport cycle in Spns transporters requires the identification of the conformational states involved in binding, translocation, and exit of the substrate and elucidation of the sequence and structural elements (e.g., protonation switches) controlling the isomerization of the transporter between functional states. Considering that the proton gradient is inwardly directed in cells, *Hn*Spns could either function as a symporter, which enables the uptake of its substrate along with protons, or, similar to Spns2, as an efflux antiporter pumping its substrate in the opposite direction. Thus, to define the proton and ligand dependence of *Hn*Spns, which presumably couples inward proton translocation to substrate extrusion, DEER distance distributions were determined at pH 4 to mimic a protonated state, at neutral pH of 7.5, and at pH 9 to favor deprotonation. For select distance pairs reporting on the relative arrangements of the substrate binding cavity and gating helices, DEER measurements were performed in the presence of potential substrates. The measurements were repeated in both lauryl maltose neopentyl glycol (LMNG) micelles and *Hn*Spns reconstituted into nanodiscs composed of plasma membrane phospholipids (Supplementary Fig. 1a and b).

### Structural integrity of *Hn*Spns mutants

To apply DEER spectroscopy, cysteine mutations were introduced at sites of interest on a cysteine-less (CL) background. The effect of introducing spin labels for each mutant on structural integrity was monitored by thermal stability analysis using circular dichroism (CD) in micelles and nanodiscs and compared to the CL *Hn*Spns (Supplementary Fig. 1 and Supplementary Table 1)[41]. Introducing spin labels did not significantly affect the structure of the CL transporter. Among protonation-mimetic mutations, D41N/E129Q and E129Q mutation alone have the highest and the lowest $T_m$ values in either micelles or lipid nanodiscs, respectively (Supplementary Fig. 1c, d and Supplementary Table 1). So, introducing the D41N mutation combined with E129Q enhances the thermal stability of *Hn*Spns. D142N mutation has a similar effect in nanodiscs. D142N putatively stabilizes the IF conformation by interrupting the structural motif A-associated charge-relay network[20]. Thus, these protonation-mimetic mutations differentially influence the thermal stability and potentially structural dynamics of *Hn*Spns.

### Proton binding closes the intracellular side

For distance pairs between the symmetry-related gating helices 2 and 11 or 5 and 8 (Fig. 2), which are directly involved in inter-domain

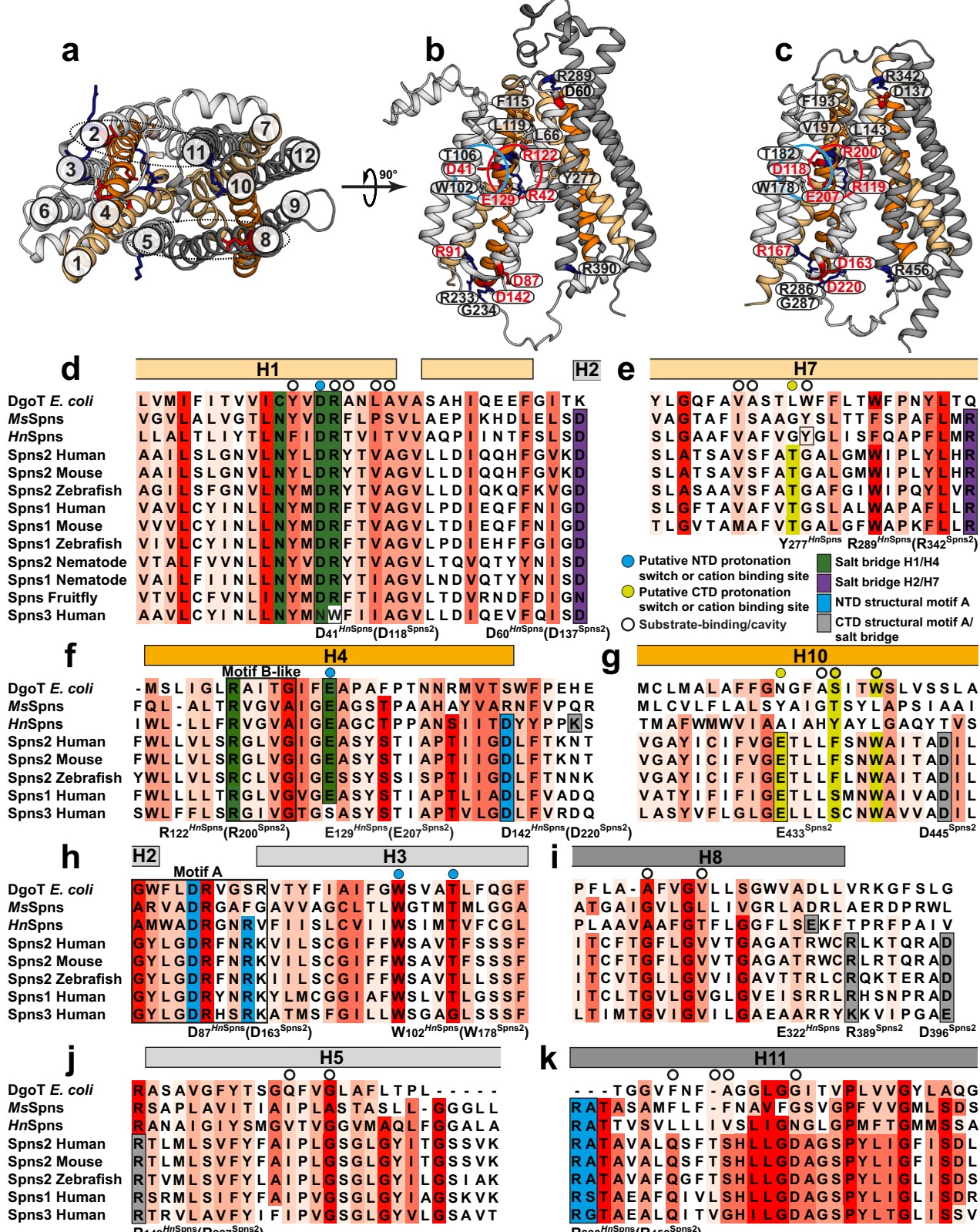

**Fig. 1 | The architecture and sequence alignment of Spns proteins.**
**a** Cytoplasmic view of the *Hn*Spns crystal structure (Protein Data Bank (PDB) code 6E9C) in the IF state with the NTD and CTD colored in light and dark gray, respectively. **b** Membrane view of *Hn*Spns with the positions of conserved and functional residues labeled. Symmetry-related TM helices 1/7 and 4/10 are colored with light and dark orange, respectively. **c** AlphaFold model of human Spns2.

**d–k** Sequence alignment of Spns proteins and their bacterial homologs, the *M. smegmatis* MDR transporter (*Ms*Spns), and the *E. coli* MFS homolog DgoT. The putative substrate and cation binding site residues are marked by circles. Residues are colored according to sequence conservation with different shades of red. Functional residues are colored differently.

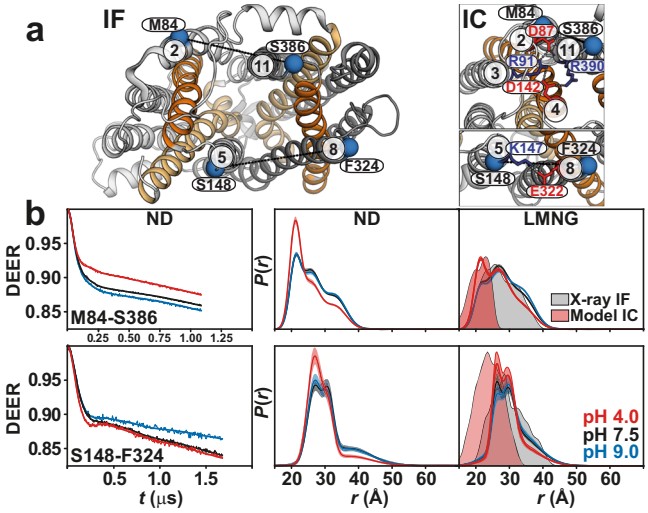

**Fig. 2 | Proton binding closes the intracellular side of *Hn*Spns. a** Spin label pairs (blue spheres) for DEER distance measurement of the gating helices across the NTD and CTD on the intracellular side of the IF crystal structure and the intracellular-closed (IC) model. **b** Raw DEER decays and fits (left) are presented for the experimentally-determined distributions $P(r)$ in nanodiscs (middle) and LMNG micelles (right) for two distance pairs, M84-S386 and S148-F324 (dotted lines in **a**), under acidic, neutral, and basic pH conditions. Confidence bands (2σ) are shown about the best fit lines. This band, which depicts the estimated uncertainty in $P(r)$, reflects the error associated with the fitting of the primary DEER trace. The distributions predicted from the IF crystal structure and the IC model are shaded gray and red, respectively.

conformational changes of *Hn*Spns, shifts to shorter distances were observed on the intracellular side at pH 4. The distance population for the TM2/TM11 pair corresponds to the formation of the structural motif A between TMs 2, 4, and 11 in an intracellular closed (IC) homology model of *Hn*Spns based on *E. coli* YajR (PDB code 3WDO), hence closing of the intracellular side. Though TM5 and TM8 move closer to each other at pH 4, the experimental distance population does not match the predicted distance corresponding to the formation of a salt bridge between these helices in the IC model. Also, in line with the DEER measurement, the formation of this salt bridge was not observed in our MD simulations. The experimental distance distributions between the gating helices in micelles at basic pH values correspond well with the distributions calculated from the X-ray IF structure (Fig. 2b). Conversely, the proton-dependent pattern of distance changes between pairs involving the cavity helices (Fig. 3) is entirely different from the gating helices, implying the involvement of their rotation and/or bending motions in the global structural transition of the protein. Notably, unlike the gating helices, the distance distributions of the cavity helices showed disagreements with the distributions predicted from the X-ray structure. Overall, the populated states at different pH values were similar for LMNG micelles and lipid nanodiscs (Figs. 2 and 3 and Supplementary Fig. 2), although the pH 4 conformational state was more populated in the lipid nanodiscs compared with micelles.

### Protonation induces a flexible outward-closed conformation
On the extracellular side, gating TM pairs 2/11 and 5/8 behave differently under acidic conditions (Fig. 4). The distance between TM2 and TM11 is shorter at pH 4. Unlike this pair, which is more structured, the TM5/TM8 pair is highly dynamic and shifts to longer distances at pH 4. The structural integrity of this spin-labeled pair is not affected compared to CL *Hn*Spns (Supplementary Table 1). The periplasmic halves of TMs 5 and 8 are intrinsically flexible due to the abundance of highly conserved glycine residues (Fig. 1i, j), implying a role in the gating

mechanism. A highly conserved periplasmic salt bridge in Spns proteins (between Asp60$^{TM2}$ and Arg289$^{TM7}$ in *Hn*Spns) tightly couples TMs 1/2 and 7 and possibly inhibits the periplasmic opening of *Hn*Spns, thus stabilizing the IF and outward-closed conformations. Indeed, we find that pH-dependent distance changes are significantly attenuated for distance pairs between TM7 and TM1/TM2 or other helices like TM3 (Fig. 5). Distance distributions for pairs involving Ser57 on TMs 1/2 connecting loop exhibit substantial proton dependence, hinting at twisting of TM1 away from TM3. Overall, the pH-dependent distance changes on the periplasmic side provide evidence for the closing of that side under acidic conditions with a flexible lateral opening toward the membrane to enable the release of lipophilic substrates such as drugs. The experimental distance distributions in both micelles and lipid nanodiscs under basic conditions correspond to the distributions predicted from the X-ray IF structure (Figs. 4 and 5 and Supplementary Fig. 3).

### Substrate binding does not drive the alternating access
*Hn*Spns was captured in the same IF conformation in the presence and absence of an uncharacterized substrate[25]. Although S1P is not a native substrate of *Hn*Spns, the conserved substrate binding residues in the Spns proteins (Fig. 1) imply a lipophilic nature for the substrate. Assuming S1P binding, its addition to *Hn*Spns in nanodiscs did not produce a substantial change in the conformational equilibrium (Supplementary Fig. 4a). The structurally unrelated antimicrobial compounds ethidium bromide and capreomycin (a cyclic polypeptide antituberculosis compound) are substrates of the *Mycobacterium smegmatis* MDR efflux transporter MSMEG_3705 (*Ms*Spns), a close homolog of Spns proteins (Fig. 1)[42]. *Ms*Spns is unable to confer resistance against the highly lipophilic drug rifampicin, the most effective, first-line antibiotic against tuberculosis. Using a cell growth assay in *Escherichia coli*[43], we observe resistance to capreomycin and ethidium bromide when cells are transformed with *Ms*Spns, over vector-transformed cells (Supplementary Fig. 5 and Tables 2 and 3). Using the distance pair S25-S386 (TM1-TM10/11) on the intracellular side of *Hn*Spns, no significant substrate-induced shift in equilibria was observed for capreomycin and ethidium bromide (Supplementary Fig. 4b, c). Interestingly, some lipophilic drugs and compounds including rifampicin, Hoechst 33342, and verapamil, shift the equilibrium towards the pH 9 state (blue arrow), inhibiting the closing of the intracellular side. Accordingly, *Ms*Spns is unable to confer resistance against rifampicin[42] (Supplementary Fig. 5). Only acidic condition or protonation-mimetic mutations triggers the conformational transition to the inward-closed state (Supplementary Fig. 4b, c). This implies that proton binding is the main driver of the alternating access mechanism in these transporters.

### Protonation of Asp41 regulates the intracellular master switch Glu129
The structure of *Hn*Spns contains two buried titratable residues within the NTD, Glu129$^{TM4}$, and Asp41$^{TM1}$. Glu129 has been hypothesized to be the master proton-coupled switch that triggers the conformational transition[25]. Due to the proximity of Glu129 and Asp41, using MD simulations, we explored the local conformational changes in this area using different protonation states of these acidic residues (all four different permutations, each in four independent 500 ns simulations) and their interactions with two nearby basic residues, substrate-binding Arg42$^{TM1}$, and Arg122$^{TM4}$. Arg122$^{TM4}$ is expected to promote deprotonation of nearby acidic residues in the IF state. During the simulations, we monitored the formation of two specific salt bridges, Glu129:Arg42 and Asp41:Arg122. Stable interactions are formed only when both acidic residues are deprotonated (Fig. 6c and Supplementary Fig. 6c). Notably, upon Asp41 protonation alone, its interaction with Arg122 is lost (Fig. 6a and Supplementary Fig. 6a), while the deprotonated Glu129-Arg42 interaction remains intact. Double

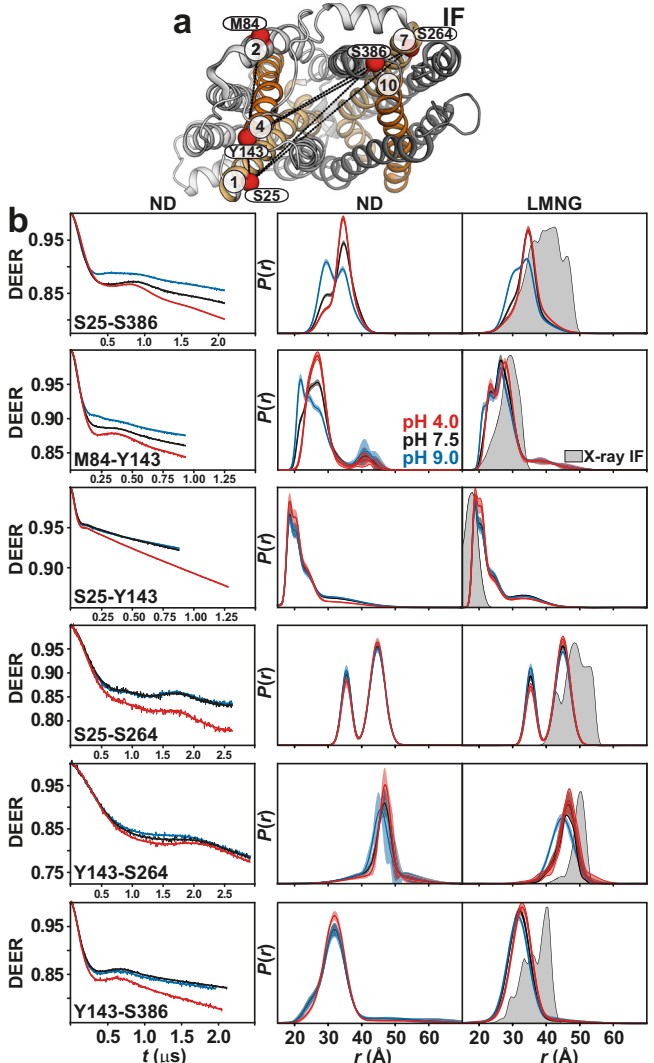

**Fig. 3 | Protonation promotes distinct cytoplasmic rearrangements of cavity helices. a** Spin label pairs reporting on the cavity helices are depicted on the intracellular side of the IF crystal structure. **b** Raw DEER decays and fits (left) are presented for the distance distributions $P(r)$ in nanodiscs (middle) and in LMNG micelles (right). The distributions predicted from the IF crystal structure are shaded gray.

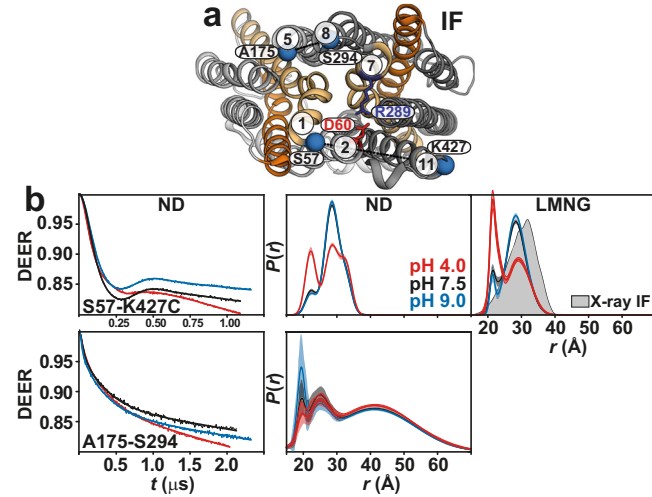

**Fig. 4 | Enhanced periplasmic flexibility of *Hn*Spns by protonation, with a lateral opening between TMs 5 and 8. a** Spin label pairs (blue spheres) on the gating helices across the NTD and CTD are depicted on the periplasmic side. **b** Raw DEER decays and fits (left) are presented for the distance distributions $P(r)$ in lipid nanodiscs (middle) and in LMNG micelles (right). The distributions predicted from the IF crystal structure are shaded gray.

proposed coupling mechanism and transport model (see Discussion), are reproducible between MD simulation replicates.

### Proton-dependent changes preserve the extracellular-closed state

In our MD simulations, we observe that protonation of Asp41 releases this residue from Arg122, enabling the rotation/bending of TMs 1 and 4 (Fig. 6a vs. 6c, middle panel). Under this condition, the charge-relay network between Asp142$^{TM4}$-Arg91$^{TM3}$-Asp87$^{TM2}$ in the NTD, essential for the assembly of the structural motif A and for the conformational transition, cannot be formed, thereby stabilizing the IF conformation (Fig. 6a, lower panel). However, protonation of Glu129 alone (Fig. 6d) triggers the largest conformational changes in TMs 1 and 4 and enables the proper formation of the charge-relay network (Fig. 6d, lower panel). Bending of the gating helix TM2 on the intracellular side occurs in response to either of these protonation events, providing a passage for substrate binding. However, only in the singly protonated Glu129 the now freed Arg42$^{TM1}$ and Tyr277$^{TM7}$ (GY$^{277}$G) become available for substrate binding (Fig. 6d). Double protonation (Fig. 6b) has a similar effect on the charge-relay network formation as the single protonation of Asp41 (Fig. 6a, b, lower panels). Interestingly, all these local and global proton-dependent conformational changes triggering the closing of the intracellular side could occur without disrupting the periplasmic Asp60$^{TM2}$-Arg289$^{TM7}$ salt bridge, which stabilizes the extracellular-closed conformation, implying the presence of the IF and O states.

### Proton switches mainly control changes on the intracellular side

Our DEER data on protonation-mimetic mutations (four different permutations) at basic pH values (e.g., pH 9.0) support the results of the MD simulations (Fig. 7 and Supplementary Figs. 7 and 8). Under basic pH conditions, proton switches other than the mutated residues are expected to be mainly deprotonated, facilitating the correspondence of the DEER data (Fig. 7) and MD simulations (Fig. 6). However, equilibrium MD simulations (Fig. 6) probe the local protonation-dependent charge interactions and conformational changes (e.g., formation of the NTD charge-relay network [lower panels], rotation/bending of TM helices [middle panels]). The ensuing global changes (e.g., closing of the intracellular side) are probed using DEER distance

protonation of these acidic residues stabilizes a distinct conformation (Fig. 6b and Supplementary Fig. 6b) in which the side chain of Arg122 continuously interacts with the backbone oxygen of Arg42 but only transiently with the backbone oxygen of Asp41 (Supplementary Fig. 6e), thereby restricting the rotational degree of freedom of TM1, which is essential for both substrate binding and the global structural transition. This might explain the higher thermal stability of the D41N/E129Q mutant (Supplementary Fig. 1c and d). In contrast, only when Asp41 is deprotonated and interacts with Arg122 can Glu129 function as the intracellular master switch. Glu129 protonation liberates Arg42, allowing it to bind the substrate and trigger local and possibly global conformational changes (Fig. 6d and Supplementary Fig. 6d). As a plausible coupling mechanism, protonation of Glu129 is promoted by decreasing the nearby positive electrostatic potential, either by compensating the positive charge of the conserved Arg122, e.g., by formation of a charge-charge interaction with deprotonated Asp41, or by binding of the negatively charged substrate to Arg42. Despite some variation due to the stochastic nature of the interactions (Supplementary Fig. 6), these main protonation-dependent charge interactions (Glu129:Arg42 and Asp41:Arg122), which are at the center of the

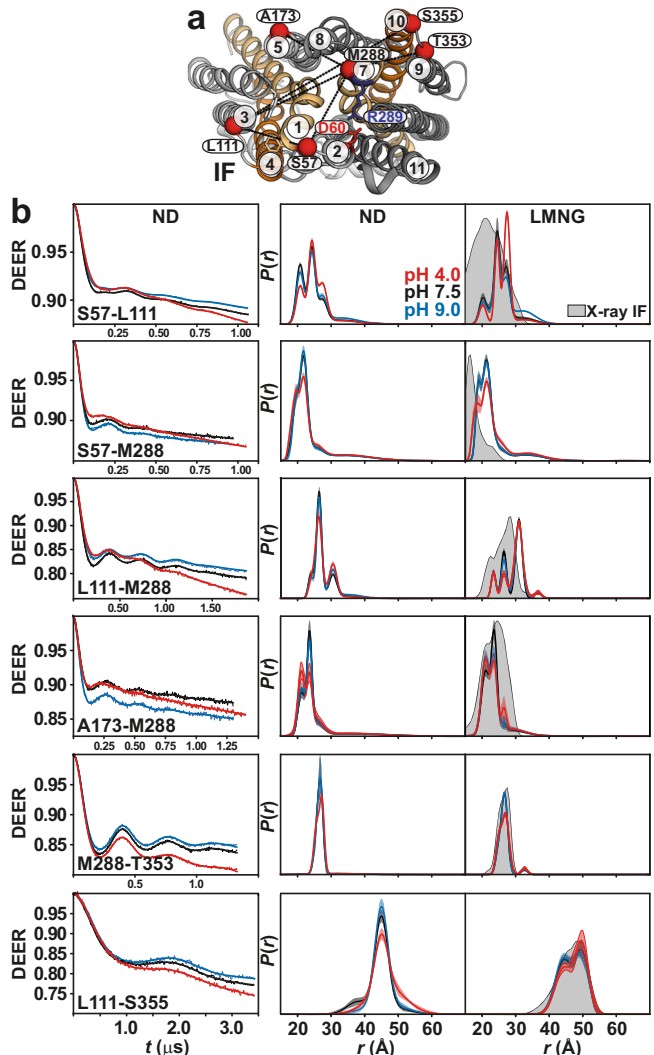

**Fig. 5 | Proton binding stabilizes an outward-closed conformation. a** Spin label pairs (red spheres) are depicted on the periplasmic side of the IF crystal structure. A salt bridge between Asp60 and Arg289 keeps the transporter in a closed conformation. **b** Raw DEER decays and fits (left) are presented for the distance distributions $P(r)$ in lipid nanodiscs (middle) and in LMNG micelles (right). The distributions predicted from the IF crystal structure are shaded gray.

pairs between NTD and CTD (e.g., M84-S386, S25-S386; Fig. 7a). On the intracellular side and probed by the M84-S386 distance between TMs 2 and 11 (Fig. 7a), DEER on the WT protein at pH 9 indicated that both intracellular-closed and open populations are sampled, and the equilibrium shifts to the intracellular-closed state at acidic pH values (red arrow). The MD simulations on the doubly deprotonated state (Fig. 6c) reveal that under this condition, the charge-relay network could form [lower panel] and therefore sampling the intracellular-closed conformation is still possible. Accordingly, in two of the four independent simulation copies of the doubly deprotonated state (Supplementary Fig. 6c), we observe sampling of the dissociated Glu129:Arg42 interaction, although, in all four copies, the Asp41:Arg122 salt-bridge remains intact. The E129Q mutation shifts the equilibrium towards the pH 4 state, which represents the formation of the charge-relay network (Fig. 6d, lower panel) and structural motif A, thus closing the intracellular side. For most of the probed intracellular pairs, the E129Q mutation captures the transporter in the pH 4 state, i.e., the inward-closed conformation (Fig. 7a and Supplementary Fig. 9). Thus, our data strongly point to E129 as the intracellular protonation master switch regulating the conformational transition between IF and inward-closed

states. However, the D41N mutation has the opposite effect, shifting the equilibrium to the pH 9 state (blue arrow in Fig. 7a and Supplementary Fig. 7). Mutation of the Asp142[TM4] in the charge-relay network produces a similar effect as the D41N mutation (Fig. 7a and Supplementary Fig. 7), as shown previously in other MFS transporters[20]. Importantly, and in line with MD simulations (Fig. 6b, lower panel), for the M84-S386 pair with the double E129Q/D41N mutation, the effect of D41N on the formation of the charge-relay network and closing of the intracellular side dominates that of E129Q. Interestingly, for the S25-S386 distance pair, which monitors the conformational changes in TM1, the reverse effect of the D41N mutation compared with the E129Q can be observed in both DEER and CW EPR data (Fig. 7a). The change in the CW spectra hints at the rotation/bending of TM1 which affects the local spin dynamics. Our MD simulations indicate that rotation/bending of TM1 could still occur in the doubly protonated state (Fig. 6b, middle panel), a prediction supported by our DEER measurements on the S25-S386 pair with D41N/E129Q mutations. On the periplasmic side, only slight and negligible shifts in the conformational equilibria can be observed in response to different protonation-mimetic mutations, indicating these proton switches mainly control conformational changes on the intracellular side of the transporter (Fig. 7b and Supplementary Figs. 8 and 10).

**Periplasmic side of *Hn*Spns senses the proton differently**

The pH-dependent conformational equilibrium of ligand-free *Hn*Spns in lipid nanodiscs and micelles was further investigated by monitoring different distances on the intracellular and periplasmic sides. Variation in the population of rising states as a function of pH was used to estimate the pK of the conformational transition (Fig. 8 and Supplementary Figs. 11 and 12). A pK value of 7.6 was obtained based on the S25-S386 intracellular pair (using a nonlinear least-squares fit), which monitors the relative movement of TM1 and TM10/TM11 (Fig. 8a, b). Since the pK reflects the protonation/ deprotonation of acidic residue(s) as the driver of *Hn*Spns isomerization, this relatively high pK value is related most likely to the protonation state of buried sites in the central cavity (i.e., Asp41[TM1] and Glu129[TM4]). The determined Hill coefficient of 1.0 hints at independent protonation events. On the contrary, an acidic pK value of $4.8 \pm 0.1$ was determined based on the pH-dependent distance changes of periplasmic pairs S57-K427 (TM1/2-TM11) and S57-L111 (TM1/2-TM3; Fig. 8a, b). This substantially lower pK on the periplasmic side could reflect the protonation state of Asp60[TM2] that forms a highly conserved salt bridge with Arg289[TM7], postulated to inhibit the periplasmic opening of *Hn*Spns. This finding further highlights the lack of correlation between the intracellular and periplasmic proton-dependent conformational changes, which was further investigated by repeating titration experiments on double and single protonation-mimetic mutations of Asp41 and Glu129 (Fig. 8c, d and Supplementary Fig. 11). We were not able to study Asp60 or Arg289 mutations due to negligible protein expression. As predicted, the D41N mutation on the intracellular distance pair shifts the pK value from 7.6 to 6.4, possibly reflecting a lower pK of the less buried residue Glu129. The D41N-E129Q double mutation completely abrogates the proton-dependent conformational changes on the intracellular side, further demonstrating that the observed pK value reflects the protonation/deprotonation of these two intracellular switches. Predictably, similar protonation-mimetic mutations have no significant effect on the extracellular pK of conformational transition (Fig. 8c, d and Supplementary Fig. 11). The pH dependence of the CW EPR spectra of the S25-S386 (TM1-TM10/11) intracellular pair reflects the change in the local spin label dynamics, possibly rotation/ bending rearrangements, and correlates with the corresponding DEER titration experiment (Supplementary Fig. 13a). Interestingly, the D41N and E129Q mutations shift the equilibrium in opposite directions towards basic and acidic pH states, respectively

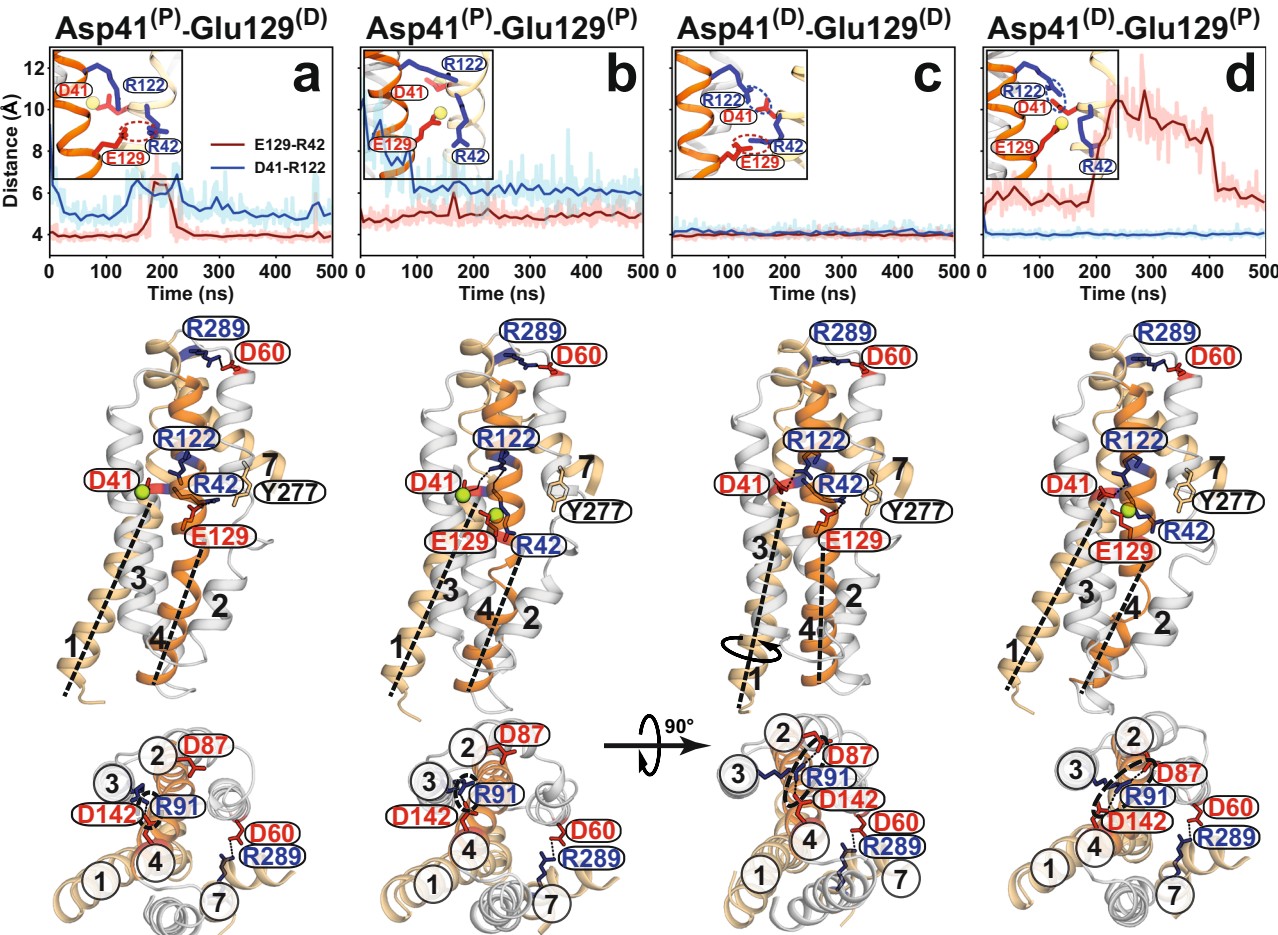

**Fig. 6 | Proton dependence of luminal salt bridges in _Hn_Spns involving substrate binding residue Arg42.** Time series (light color) and moving averages (dark color) of Glu129-Arg42 (red) and Asp41-Arg122 (blue) side-chain distances from MD simulations of the IF _Hn_Spns structure in the membrane are used to monitor the stability of functionally relevant salt bridges in simulations under different titration states (**a**–**d**) of Asp41 and Glu129 (indicated by D and P for deprotonated and protonated, respectively). Data for all simulation replicas are shown in Supplementary Fig. 6. Representative snapshots of the molecular structure from each simulation condition are shown from the side (middle panels) and cytoplasmic (bottom panels) views. Only when Asp41 is deprotonated and interacts with Arg122 can Arg42 completely disengage from protonated Glu129 and bind the substrate (panel **d**). The Asp41 and Glu129 protonation-dependent conformational changes in TMs 1 and 4 could occur independently of the periplasmic salt bridge stabilizing the extracellular-closed conformation (middle panels). Upon protonation of Asp41, the charge-relay network between Asp87-Arg91-Asp142, which is essential for the conformational transition, cannot be formed, thereby stabilizing the IF conformation (**a** and **b**, lower panels).

(Supplementary Fig. 13b). Similar pK values were obtained in LMNG micelles and lipid nanodiscs (Supplementary Fig. 12).

### DEER-guided refinement of IF _Hn_Spns structure and modeling the O state

Starting from the X-ray IF conformation of _Hn_Spns (WT), we performed well-tempered metadynamics, an enhanced sampling MD technique[44], to sample different conformations of the transporter ranging from IF to O. The conformations were sampled in a phase space composed of two orientation-based collective variables (CVs) denoted as $\alpha$ and $\beta$ angles representing the intracellular and extracellular openings, respectively (Supplementary Fig. 14). During 200-ns metadynamics simulations, a putative O conformation was captured. This conformation was stabilized by two salt bridges, namely Arg289[TM7]-Asp60[TM2] and Arg390[TM11]-Asp142[TM4] (Supplementary Fig. 14e). The following 500-ns equilibrium simulation of the obtained O structure confirmed the stability of the formed salt bridges as well as the intracellular closure with reference to the starting IF conformation (Supplementary Fig. 14c). In this O state equilibrium trajectory, a pathway is possible for proton entry from the periplasmic side to Asp41 and Glu129 via the main vestibule (Supplementary

Fig. 14f). Also, a proton transfer pathway exists from the main vestibule to the acidic residues in the NTD charge-relay network (e.g., Asp142) that stabilizes the intracellular-closed conformation. The main vestibule in the occluded state could accommodate the substrate (Supplementary Fig. 14f). Both the IF and O structures were refined using the DEER distance histograms at pH 9 and 4, respectively, employing restrained-ensemble MD (reMD) simulation (Fig. 9)[40]. The refined models match the experimentally determined distance populations (Supplementary Figs. 15 and 16). The intracellular protonation master switch Glu129 is predicted to have a pK_a of approximately 4.0 and 6.4 (H++ server) in the IF and O models, respectively[45]. This could indicate further sequestration and protonation of this residue in the O state. Interestingly, the predicted pK_a for Glu129 in the O state matches the experimental intracellular pK of conformational changes in lipid nanodiscs for the D41N mutation which Asp41 is removed as a protonation site and the pK_a therefore is related to the Glu129 protonation reaction (Fig. 8c). With its side chain carboxyl relatively more sequestered, Asp41 is predicted to have a pK_a of 6.2 in the IF state, which is increased to 7.8 in the O state, corresponding to the increased intracellular pK for the WT protein with intact Asp41 besides Glu129 (Fig. 8a).

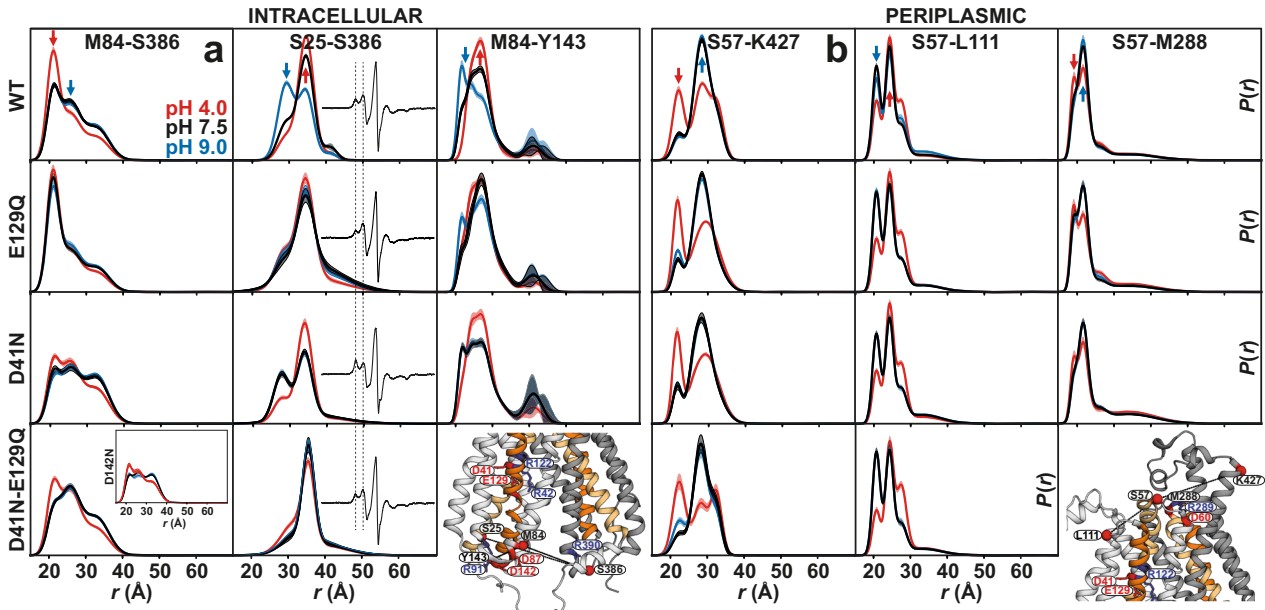

**Fig. 7 | Weak coupling of proton-dependent conformational changes on the intracellular and periplasmic sides of *Hn*Spns.** Effect of protonation-mimetic mutation of acidic residues on conformational states in equilibrium on the **a** intracellular and **b** periplasmic side (Supplementary Figs. 7–10). The single (E129Q, D41N, and D142N) or double (D41N/E129Q) mutations were combined with the double-cysteine mutations. DEER experiments in the presence of these mutations

support the results of the MD simulations and reveal that Asp41 protonation has an opposite effect from that of Glu129 on the conformational changes on the intracellular side, and the periplasmic side remains mostly invariant in response to these mutations. Confidence bands (2σ), which represent the estimated uncertainty in *P(r)*, are shown about the best fit lines. Arrows highlight the pH 4 (red) and pH 9 (blue) distance components.

The NTD charge-relay network is accompanied by an inter-domain hinge loop (i.e., L6-7). This linker peptide contains a number of polar residues (often positively charged) as well as an amphipathic α-helix (Fig. 1b). In the IF crystal structure of *Hn*Spns, the L6-7 hinge loop is tethered to TM4 and TM2 via interactions between Arg233 and Asp87[motif-A], and between Gly234 and Asp142[TM4] (Fig. 1b)[25]. In the O state (Fig. 9c), this tether is released (M84-S236 distance: pH 7.5 to 4.0; Supplementary Figs. 2 and 9), and the structural motif A is formed. According to the "motif A hypothesis"[15], proton-induced conformational rearrangements are sensed by the amphipathic α-helix, which, in turn, further induces a conformational change in L6-7. Through the charge-relay network, the positive charges (e.g., Arg233) of the loop may weaken the interdomain interaction between Asp87[motif-A] and TM11, thus destabilizing the O state. Acidic residues in the charge-relay network (e.g., Asp142) could also function as protonation switches (Fig. 7a and Supplementary Fig. 7)[20].

## Discussion

Using an integrated computational and spectroscopic approach, the extensive results presented here reveal new mechanistic details of *Hn*Spns as a close homolog of the human Spns proteins, including its conformational dynamics following protonation/deprotonation of Glu129, which is regulated by the protonation state of Asp41 and substrate binding. Our comprehensive MD simulations implicate neutralization of the highly conserved Arg122 or Arg42 in this regulatory mechanism. The results also indicate a noncanonical, ligand-dependent alternating access in the absence of an obvious OF conformational state. While our IF model is similar to the X-ray structure, we have detected and generated an occluded state of *Hn*Spns in the membrane. A periplasmic salt bridge keeps the transporter in a closed conformation, yet proton-dependent conformational dynamics are significantly enhanced on the periplasmic side (Fig. 10a) to allow ligand exchange between the lumen and the membrane/solution. Our DEER results in the presence of S1P and other potential substrates (Supplementary Fig. 4) indicate that protonation/deprotonation events, not

substrate binding, constitute the main driver of the alternating access. Another distinctive feature of *Hn*Spns is the weak coupling between proton-dependent conformational changes on the cytoplasmic and periplasmic sides that are likely facilitated by more flexible gating helices. Accordingly, the considerably lower periplasmic pK of conformational changes compared with the intracellular side implies the functional necessity of an inwardly directed proton gradient across the membrane.

The resistance assays to structurally unrelated antimicrobial compounds for the bacterial homolog *Ms*Spns suggest a previously underappreciated role of Spns proteins as broad-specificity efflux pumps[42]. However, this phenomenon remains to be tested more explicitly. Due to the similarity in binding pocket properties between bacterial homologs and human Spns proteins, the human proteins could potentially mediate clinically relevant resistance to antibiotics and chemotherapeutic drugs, a hypothesis that also requires further exploration. High Spns2 or Spns3 expression is associated with poor prognosis in chemotherapy patients with acute myeloid leukemia[8]. Notably in Spns3, the substrate binding pocket is less charged, with the substrate binding arginine in TM1 replaced with a tryptophan (Fig. 1d). Similar to other drug-proton antiporters 1 (DHA1) MFS transporters like MdfA, the observed flexibility of the protonated *Hn*Spns and a less hydrophilic binding cavity are crucial to populate transport-competent intermediates that fit diverse substrates[46–49]. As previously shown for MdfA, proton coupling can be different for neutral and charged substrates[49,50]. Notably, the protonation master switch glutamate in Spns proteins is replaced by a serine residue in human Spns3 (Fig. 1f), though glutamate is located in the middle of TM6. Interestingly, for the MDR protein LmrP, binding of a negatively charged lipid in the binding pocket facilitates substrate polyspecificity[51]. Similarly, in bacterial Spns proteins, binding of lipid to the substrate binding arginine may enhance their ability to bind a variety of neutral or even positively-charged substrates. Many MDR transporters from MATE family (e.g., pfMATE) and ABC superfamily (e.g., MsbA, ABCB1, ABCC1, and ABCG2) also physiologically function

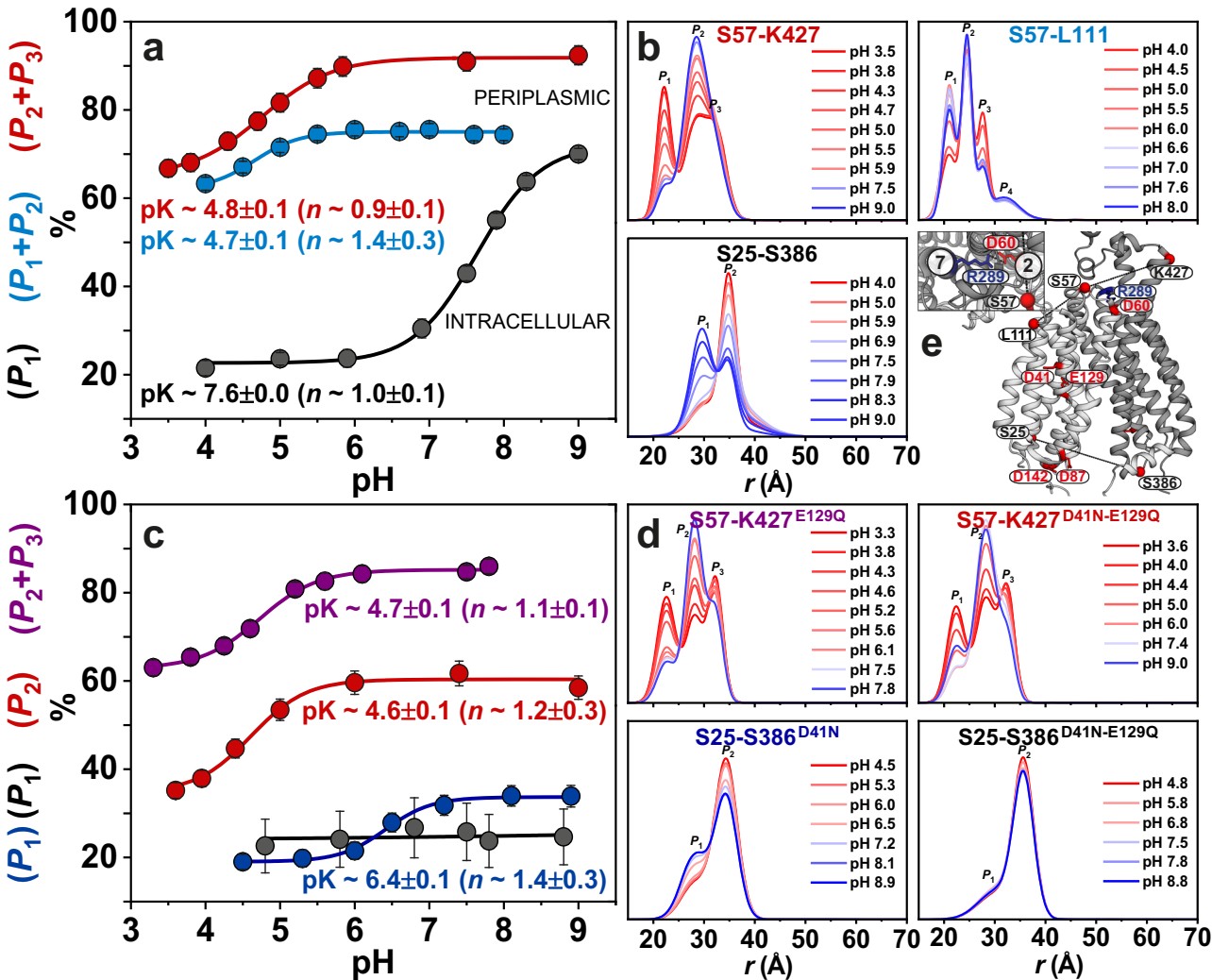

**Fig. 8 | Intracellular and periplasmic sides of *Hn*Spns sense its protonation state differently. a, b** Distance distributions of periplasmic (S57-K427 and S57-L111) and intracellular (S25-S386) pairs obtained at different pH values in lipid nanodiscs (Supplementary Fig. 11) and LMNG micelles (Supplementary Fig. 12). The variation in the population of rising or decreasing distance peaks as a function of pH was used to estimate the pK value for conformational changes in *Hn*Spns on either membrane side. **c, d** Effect of protonation-mimetic mutation of acidic residues on pK values for conformational changes. The D41N-E129Q double mutation completely abrogates conformational changes on the intracellular side while the periplasmic side remains mainly unaffected, further demonstrating the weak coupling of transmembrane proton-dependent conformational changes. The error bars (panels **a** and **c**) for the populations of distance peaks (centers: populations from the best fit distance distributions, panels **b** and **d**) represent the uncertainty in the fit parameter reported at the 2σ (95%) confidence level. Source data are available as a Source Data file.

as lipid flippases/transporters. Interestingly, ABCC1 and ABCG2 also participate in S1P transport[52].

From the flexible periplasmic side, protons can access Asp41 and Glu129 via the main vestibule (Fig. 10a and Supplementary Fig. 14f). Protons can also be released via water to the intracellular side from the main vestibule (Fig. 9b, c and Supplementary Fig. 14f). Both the refined IF and O models predict a putative alternative tunnel in the NTD that extends from the periplasmic surface to Asp41 (Fig. 9b, c). At the end of the tunnel, a pocket of conserved polar residues including the intracellular protonation master switch Glu129, Arg122, and the putative substrate-binding residue Arg42, lies next to Asp41[28]. The top of the tunnel may be constricted by highly conserved residues Trp206[TM6] and Phe56[TM1/EH1-2] (Fig. 9d). Based on our DEER distance of the S57-L111 pair as a reporter (TM1-TM3; Supplementary Fig. 9), we predict these residues act as gatekeepers that sense the proton-dependent structural rearrangements on the extracellular but not on the intracellular side. Conservation analysis[53] outlines a putative proton transfer pathway to Asp41, including conserved residues along TM3 (e.g., Trp102, Ser103, Thr106, Cys109

and Gly110) and TM6 (Trp206, Arg207, Val209, Phe210, Gly214 and Gly217).

Based on the DEER-refined IF and O structures of *Hn*Spns in lipid membranes and its local and global proton-dependent conformational dynamics, we propose an antiport model for *Hn*Spns. This model identifies residues critical for protonation and its regulation, reveals how sequential protonation of these switches drives the conformational transitions and suggests a mechanism of energy coupling to substrate transport as well as a putative path for proton translocation (Figs. 9 and 10). In the model, the stable resting state is IF (Fig. 10b, steps 2 and 3), whereas the O (steps 1 and 4) represents a higher-energy state. In the transition from O to IF (Fig. 10b, step 1 to 2), Asp41 protonation, possibly through the putative periplasmic tunnel, promotes Glu129 deprotonation and its neutralization by the formation of a charge pair with Arg42. Subsequently, proton transfer to the acidic residues of the intracellular charge-relay network (e.g., Asp142), and its disruption stabilize the IF conformation[20] priming it for substrate binding from the intracellular side (step 2). Proton transfer from Asp41 to Glu129 liberates Arg42 to bind substrates such as S1P (step 2 to 3).

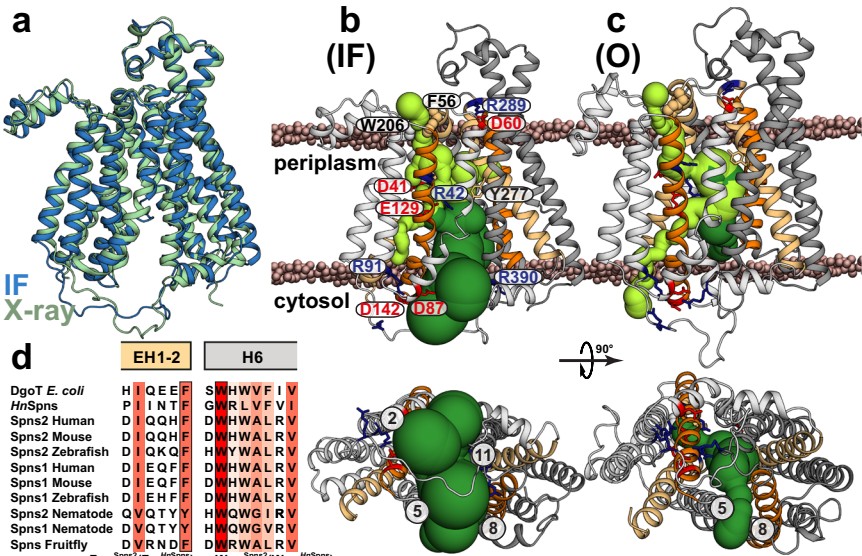

**Fig. 9 | Sampled conformational states of *Hn*Spns reveal a noncanonical, ligand-dependent alternating access in the absence of an obvious outward-facing state. a** The refined X-ray structure of *Hn*Spns in the lipid membrane based on the pH 9 DEER distance populations in nanodiscs (Supplementary Fig. 15), is IF and comparable to the IF X-ray structure. **b** Calculated tunnels using software MOLEonline 2.5 within the IF model (dark green) indicate that the substrate can reach the binding site either from the inner leaflet of the membrane through TMs 2/11 and 5/8 gates or directly from the cytoplasm. Thus, both lipophilic and water-soluble substrates can get access to the transporter. In this conformation, Arg42$^{TM1}$

and Tyr277$^{TM7}$ in the binding site are poised to bind the substrate. **c** Due to the formation of the structural motif A, the intracellular gate between TMs 2 and 11 is closed in the modeled doubly-occluded state based on the pH 4 DEER distances (Supplementary Fig. 16). However, the gate between TMs 5/8 is not completely occluded and is still accessible from the membrane. The periplasmic salt bridge Asp60$^{TM2}$–Arg289$^{TM7}$ is also intact in this O state. **d** A putative periplasmic tunnel (light green) for proton translocation is predicted in both refined models of IF and O. The top of the tunnel may be constricted by highly conserved residues Trp206$^{TM6}$ and Phe56$^{TM1/EH1-2}$.

Glu129 protonation triggers the closing of the intracellular side and transition to a periplasmic flexible but closed conformation (step 3 to 4). This enables the substrate release to the periplasmic leaflet, possibly through the putative periplasmic gate between TMs 5/8 (Fig. 10a) and restoration of the Arg42-Glu129 charge pair and Glu129 deprotonation. For an electrogenic substrate/nH+ (with n ≥ 2) transport[28], Glu129 is possibly re-protonated from the flexible periplasmic side (step 4 to 1). Otherwise, proton transfer from Glu129 to the intracellular charge-relay network and protonation of Asp41 in step 4 are sufficient to directly reset the transporter to the IF conformation (step 4 to 2).

The proposed transport model highlights distinct aspects of alternating access in Spns proteins. Similarities can also be detected to the proposed alternating access mechanisms in other MFS transporters, including proton-coupled sugar acid and sugar symporters such as *E. coli* DgoT and XylE, and MDR efflux antiporters like MdfA and LmrP[27,28,46,51]. A similar regulatory effect of Asp41 protonation in *Hn*Spns is proposed for Asp27 in XylE, and several of the conserved functional residues in the NTD of Spns proteins are also present in DgoT (Fig. 1d, f). Likewise, active transport in DgoT requires reversible protonation of both Asp46 and Glu133, residues corresponding to the protonation switches in Spns proteins[28]. A recent study of MdfA in nanodiscs suggested noncanonical alternating access involving a flexible O conformation, similar to the O states we have detected in *Hn*Spns[46].

Among human Spns transporters, lysosomal Spns1 most likely functions as a symporter, removing degradation products, including sphingolipids, from the lysosomal lumen by coupling to the proton gradient across the lysosomal membrane. In contrast, Spns2 is predicted to function as an S1P efflux antiporter, given the inwardly directed H+ gradient across the cell membrane. Extracellular acidosis that is typically found in inflammatory tissues or tumors might stimulate the Spns2 activity and enhance S1P export. This strongly suggests that identical functional elements (e.g., protonation switches;

Fig. 1) can support substrate transport in opposite directions. Using a similar integrated approach to define the transport mechanism of other Spns family members and their prokaryotic homologs will identify the key commonalities and differences in their mechanisms, highlighting the mechanistic flexibility that enables their diverse function with transformative therapeutic potential.

## Methods

No statistical methods were used to predetermine sample size. The experiments were not randomized. The investigators were not blinded to allocation during experiments and outcome assessment.

### Site-directed mutagenesis

Codon-optimized *Hn*Spns (GenScript) was cloned into pET19b vector encoding an N-terminal 10-His tag under control of an inducible T7 promoter. The five cysteine residues in *Hn*Spns were mutated (C98A, C109S, C132S, C442A, C457A) via site-directed mutagenesis with complementary oligonucleotide primers, yielding the CL protein. This construct was used as the template to introduce double-cysteine pairs and background mutations. Substitution mutations were generated using a single-step PCR in which the entire template plasmid was replicated from a single mutagenic primer. *Hn*Spns mutants were sequenced using both T7 forward and reverse primers to confirm mutagenesis and the absence of aberrant changes. Mutants are identified by the native residue and primary sequence position followed by the mutant residue. Codon-optimized *Ms*Spns (GenScript) was cloned into pET19b vector encoding an N-terminal 10-His tag under control of an inducible T7 promoter.

### Expression, purification, and labeling of *Hn*Spns

*Hn*Spns was expressed and purified using a similar protocol as previously published[54]. *Escherichia coli* C43 (DE3) cells (Sigma-Aldrich) were freshly transformed with pET19b vector encoding recombinant *Hn*Spns mutants. A transformant colony was used to inoculate

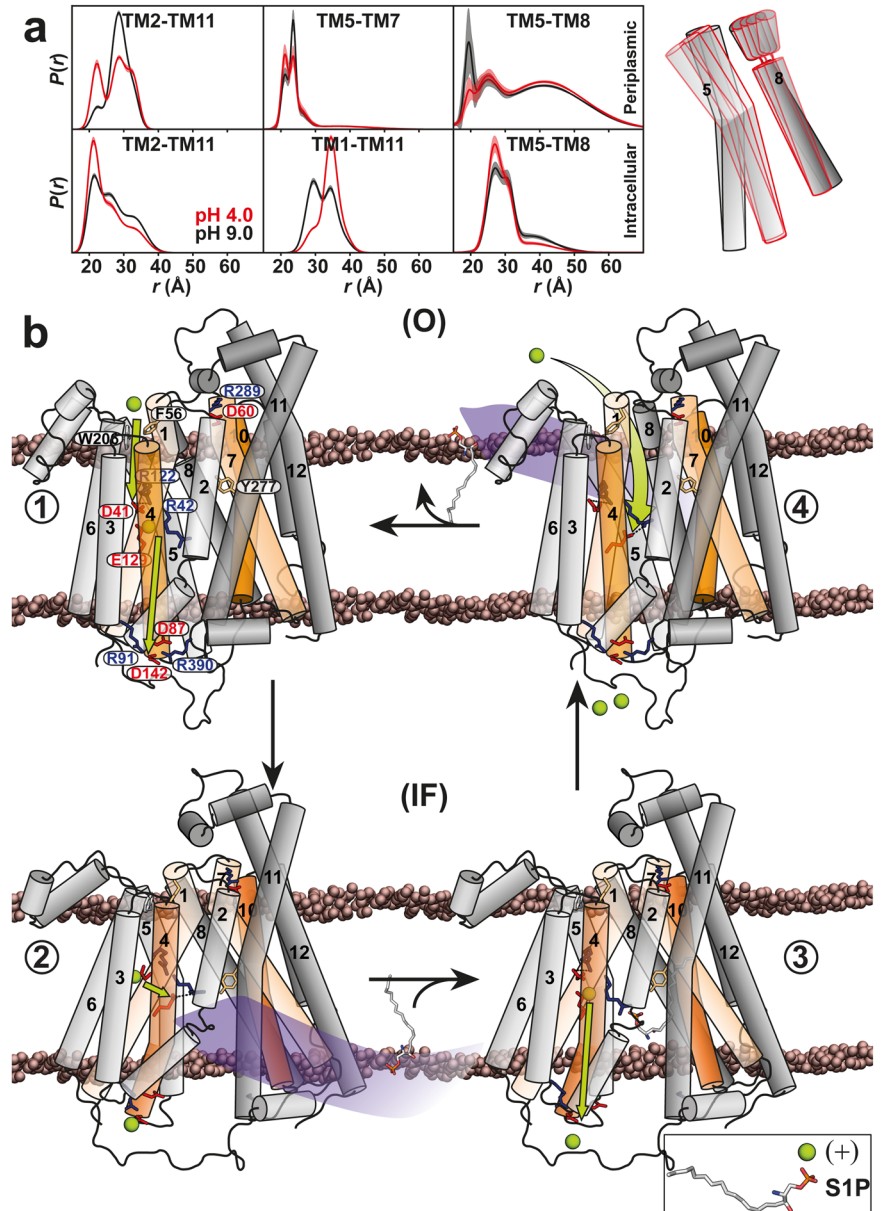

**Fig. 10 | Proposed model of S1P/H⁺ antiport in *Hn*Spns. a** Proton-dependent conformational dynamics are significantly enhanced on the periplasmic side, providing a pathway for ligand exchange. **b** In this model, the stable resting state is IF, and O is of relatively high energy. Side chains of the functionally conserved residues are represented as sticks. Asp41 protonation promotes the Glu129 deprotonation and formation of a charge pair with Arg42 (**1**). Subsequently, the released proton translocates to the intracellular side, where the acidic residues of the charge-relay network (e.g., Asp142) are protonated, stabilizing the IF conformation for substrate binding from the intracellular side (**2**). Proton transfer from Asp41 to Glu129 liberates Arg42 to bind the substrate, e.g., S1P (**3**). Glu129 protonation triggers closing of the intracellular side and transition to a periplasmic flexible but closed conformation (**4**). This enables the substrate release to the periplasmic leaflet, possibly through the putative periplasmic gate between TMs 5/8, restoration of the Arg42-Glu129 charge pair, and Glu129 deprotonation. For an electrogenic transport, Glu129 is re-protonated from the flexible periplasmic side (**4** to **1**). Otherwise, proton transfer from Glu129 to the intracellular charge-relay network and protonation of Asp41 in step 4 are sufficient to directly reset the transporter to the IF conformation (**4** to **2**).

Luria–Bertani (LB) media (Fisher Bioreagents) containing 0.1 mg/mL ampicillin (Gold Biotechnology), which was grown overnight (~15 h) at 34 °C and was subsequently used to inoculate 3–6 L of minimal medium A at a 1:50 dilution. Cultures were incubated while being shaken at 37 °C until they reached an absorbance at 600 nm (Abs₆₀₀ₙₘ) of ~0.8, at which time *Hn*Spns expression was induced by the addition of 1 mM IPTG (Gold Biotechnology). The cultures were incubated overnight (~15 h) at 20 °C and then harvested by centrifugation. Cell pellets were resuspended in resuspension buffer (20 mM Tris·HCl, pH 8.0, 20 mM NaCl, 30 mM imidazole, and 10% [vol/vol] glycerol) at 15 mL per liter of culture, including 10 mM DTT and 1 mM PMSF, and the cell suspension was lysed via sonication. Cell debris was removed by centrifugation at

9,000 × *g* for 10 min. Membranes were isolated from the supernatant by centrifugation at 200,000 × *g* for 1.5 h.

Membrane pellets were solubilized in resuspension buffer containing 5 mM LMNG (Anatrace) and 0.5 mM DTT and incubated on ice with stirring for 1 h. Insoluble material was cleared by centrifugation at 200,000 × *g* for 30 min. The cleared extract was bound to 1.0 mL (bed volume) of Ni-NTA Superflow resin (Qiagen) at 4 °C for 2 h. After washing with 10 bed volumes of buffer containing 30 mM imidazole, *Hn*Spns was eluted with buffer containing 300 mM imidazole.

Double-cysteine mutants were labeled with two rounds of 20-fold molar excess 1-oxyl-2,2,5,5-tetramethylpyrroline-3-methyl methanethiosulfonate (Enzo Life Sciences) per cysteine on ice in the dark

over a 4-h period, after which the sample was kept on ice at 4 °C overnight (~15 h) to yield the spin label side chain R1. Unreacted spin label was removed by size exclusion chromatography over a Super-dex200 Increase 10/300 GL column (GE Healthcare) into 50 mM Tris/MES, pH 7.5, 0.1 mM LMNG, and 10% (vol/vol) glycerol buffer. Peak fractions of purified $Hn$Spns were combined and concentrated using an Amicon Ultra 100,000 MWCO filter concentrator (Millipore), and the final concentration was determined by $A_{280}$ measurement ($\varepsilon = 95{,}340$ M$^{-1}$·cm$^{-1}$) for use in subsequent studies.

## Reconstitution of $Hn$Spns into nanodiscs

1-palmitoyl-2-oleoyl-sn-glycero-3-phosphocholine (POPC), 1-palmitoyl-2-oleoyl-sn-glycero-3-phosphoethanolamine (POPE), 1-palmitoyl-2-oleoyl-sn-glycero-3-phospho-L-serine (POPS) and L-α-phosphatidylinositol (Liver, Bovine, POPI) (Avanti Polar Lipids) were combined in a 17.5:44:27.5:11 (mol/mol) ratio, dissolved in chloroform, evaporated to dryness on a rotary evaporator, and desiccated overnight under vacuum in the dark. The lipids were hydrated in 50 mM Tris/MES, pH 7.5, buffer to a final concentration of 20 mM, homogenized by freezing and thawing for 10 cycles, and stored in small aliquots at −80 °C. MSP1D1E3 was expressed and purified as previously described[55] and dialyzed into 50 mM Tris/MES, pH 7.5, buffer. MSP1D1E3 was concentrated using a 10,000 MWCO filter concentrator and the final protein concentration was determined by $A_{280}$ measurement ($\varepsilon = 29{,}910$ M$^{-1}$·cm$^{-1}$).

For reconstitution into nanodiscs, spin-labeled double-cysteine mutants in LMNG micelles were mixed with lipid mixture, MSP1D1E3, and sodium cholate in the following molar ratios: lipid:MSP1D1E3, 50:1; MSP1D1E3:$Hn$Spns, 10:1; and LMNG + cholate:lipid, 3:1. Reconstitution reactions were mixed at 4 °C for 1 h. Detergent was removed from the reaction by addition of 0.1 g/mL Biobeads (Bio-Rad) and incubation at 4 °C for 1 h. This was followed by another addition of 0.1 g/mL Biobeads with 1-h incubation, after which 0.2 mg/mL Biobeads were added and mixed overnight. The next day, 0.2 mg/mL Biobeads were added and mixed for 1 h. The reaction was filtered using a 0.45-μm filter to remove Biobeads. Full nanodiscs were separated from empty nanodiscs by size exclusion chromatography into 50 mM Tris/MES, pH 7.5, and 10% (vol/vol) glycerol buffer. The $Hn$Spns-containing nanodiscs were concentrated using Amicon ultra 100,000 MWCO filter concentrator, and then characterized using SDS/PAGE to verify reconstitution and estimate reconstitution efficiency. The concentration of spin-labeled mutants in nanodiscs was determined as described previously by comparing the intensity of the integrated continuous-wave electron paramagnetic resonance (CW-EPR) spectrum to that of the same mutant in detergent micelles[56].

## Thermal stability analysis

Structural integrity of the spin-labeled DEER and protonation-mimetic mutants in LMNG micelles and lipid nanodiscs were monitored using CD spectroscopy. Protein samples were diluted to 0.1 mg/mL in 10 mM sodium phosphate, pH 7.5 buffer (CD buffer) with or without 0.1 mM LMNG, for detergent micelles and nanodiscs, respectively. CD measurements were recorded on an Applied Photophysics Chirascan V100. Thermal denaturation was continuously monitored every 0.5 °C at 222 nm from 10 to 95 °C with a 0.2 °C tolerance and corrected with the same measurements using CD buffer with LMNG or empty nanodiscs. Melting temperature ($T_m$) was determined by a non-linear least squares analysis of each individual curve in OriginPro (OriginLab).

## Drug resistance assay

Resistance to toxic concentrations of capreomycin, rifampicin and ethidium bromide, conferred by $Ms$Spns, was carried out as previously described[43]. *Escherichia coli* BL21 (DE3) were transformed with empty pET19b vector and pET19b encoding $Ms$Spns WT. A dense overnight

culture from a single transformant was used to inoculate 10 mL of LB broth containing 0.1 mg/mL ampicillin to a starting $Abs_{600}$ of 0.0375. Cultures were grown to $Abs_{600}$ of 0.3 at 37 °C and expression of the encoded construct was induced with 1.0 μM IPTG (Gold Biotechnology). Expression was allowed to continue at 37 °C for 2 h, after which the $Abs_{600}$ of the cultures was adjusted to 0.5. The cells were then used to inoculate (1:20 dilution, starting $Abs_{600} = 0.025$) a sterile 96-well microplate (Greiner) containing 50% LB broth, 0.1 mg/mL ampicillin, and 2.5–10 μg/mL rifampicin or 8.8–70 μg/mL of capreomycin/ethidium bromide. Microplates were incubated at 37 °C with shaking at 250 rpm for 6 h. The cell density ($Abs_{600}$) was measured every 2 h on a SpectraMax i3 microplate reader and plotted as is or normalized to the 0 μg/mL drug well to obtain a relative absorbance, which accounts for growth behavior of the vector and WT $Ms$Spns in the absence of drug. Each data point was performed in triplicate, and the experiments were repeated twice. The $p$ values (Supplementary Tables 2 and 3) were determined by an unpaired $t$ test (GraphPad).

## CW-EPR and DEER spectroscopy

CW-EPR spectra of spin-labeled $Hn$Spns samples were collected at room temperature on a Bruker EMX spectrometer operating at X-band frequency (9.5 GHz) using 10 mW incident power and a modulation amplitude of 1.6 G. DEER spectroscopy was performed on an Elexsys E580 EPR spectrometer operating at Q-band frequency (33.9 GHz) with the dead-time free four-pulse sequence at 83 K[57]. Pulse lengths were 20 ns (π/2) and 40 ns (π) for the probe pulses and 40 ns for the pump pulse. The frequency separation was 63 MHz. To ascertain the role of H$^+$, samples were titrated to pH 4 and 9 with empirically determined amounts of 1 M citric acid and 1 M Tris, respectively, and confirmed by pH microelectrode measurement. The substrate-bound state was generated by the addition of 1 mM substrates at pH 7.5 or 9.0. Samples for DEER analysis were cryoprotected with 24% (vol/vol) glycerol and flash-frozen in liquid nitrogen.

Primary DEER decays were analyzed using a home-written software (DeerA, Dr. Richard Stein, Vanderbilt University) operating in the Matlab (MathWorks) environment as previously described[58]. Briefly, the software carries out a global analysis of the DEER decays obtained under different conditions for the same spin-labeled pair. The distance distribution is assumed to consist of a sum of Gaussians, the number and population of which are determined based on a statistical criterion. The generated confidence bands were determined from calculated uncertainties of the fit parameters. We also analyzed DEER decays individually and found that the resulting distributions agree with those obtained from global analysis. Comparison of the experimental distance distributions with the $Hn$Spns crystal structure (PDB code 6E9C) using a rotamer library approach was facilitated by the MMM 2018.2 software package[59]. Rotamer library calculations were conducted at 298 K. For a few samples, a noticeable change in the intermolecular background was observed in nanodiscs at pH 4, giving rise to a steep decay relative to higher pH. This reversible change in background is associated with reversible clustering of individual nanodisc particles without affecting the obtained DEER distance information[54].

## Molecular dynamics (system setup)

μs-scale MD simulations in multiple replicates were performed to probe the conformational response of the membrane-embedded WT protein to changes in the protonation states of the two putative proton-binding sites (Asp41 and Glu129). The crystal structure of $Hn$Spns in the IF conformation (PDB code 6E9C) was prepared for MD simulations. Missing side chains and hydrogen atoms were added using the PSFGEN plugin of VMD[60]. Neutral N-terminal and C-terminal caps were added to the beginning and end of the polypeptide chain, respectively, followed by the addition of a disulfide bond between residues Cys442 and Cys457 using PSFGEN. The protonation states of

titratable residues were estimated with PROPKA[61]. The protein was embedded in a lipid bilayer generated by CHARMM-GUI[62], followed by removing steric clashes between the protein and lipids. Orientations of Proteins in Membranes (OPM) were used to obtain the orientation of the protein in the bilayer[63]. The same lipid composition as lipid nanodiscs was employed in the MD simulations. The complex was subsequently solvated in VMD (final system size: ~233,000 atoms).

## Simulation conditions

All simulations were performed employing the fully atomistic CHARMM36m[64] and CHARMM36[65] force fields for the protein and lipids, respectively. The TIP3P model was used for water molecules[66]. The systems were simulated with NAMD[67,68] using the following parameters. For the short-range, non-bonded interactions, a 12-Å cutoff was employed, with switching starting at 10 Å. The particle mesh Ewald (PME) algorithm was employed to calculate long-range electrostatic interactions with a grid density of $1\,\text{Å}^{-1}$, and a PME interpolation order of 6. Bonds involving hydrogen atoms were kept fixed by employing the SHAKE algorithm. The temperature was sustained at 310 K using Langevin thermostat with a damping coefficient of $1.0\,\text{ps}^{-1}$. The pressure was kept at 1 atm by the Nosé-Hoover Langevin piston barostat with period and decay of 100 and 50 fs, respectively. All simulations were performed in a flexible cell allowing the dimensions of the periodic cell to change independently while keeping the cell aspect ratio in the $xy$ plane (membrane plane) constant. A 2-fs timestep was used in all simulations. Lennard-Jones and PME forces were updated at one and two timesteps, respectively. At least three different sets of simulations were conducted to investigate: 1) proton-coupled conformational response of the *Hn*Spns in the IF conformation (four replicates), 2) capturing a stable O conformation of the transporter, and 3) structural refinement of the IF and O conformations by incorporating the experimental DEER data.

## Equilibrium MD to investigate proton-coupled conformational changes

Four different systems were simulated to investigate the coupling of the two putative proton-binding sites (Asp41 and Glu129) to the conformational changes of the transmembrane region of *Hn*Spns. These four systems cover all the combinations of different protonation states of the two titratable residues. To improve statistics, each of the four systems was replicated four times and simulated for 500 ns $(4 \times 4 \times 500\,\text{ns} = 8\,\mu\text{s})$. The replicas were generated employing the Membrane Mixer Plugin (MMP)[69], avoiding any bias from the initial placement of the lipids.

## Enhanced sampling simulations capturing a stable occluded conformation

Well-tempered metadynamics[44,70] was employed to capture the O state of *Hn*Spns by sampling the conformation of the system in a phase space defined by two orientation-based collective variables (CVs), denoted as α and β, describing the cytoplasmic and extracellular opening/closing of the transmembrane region of the transporter, respectively (Supplementary Fig. 14a). The two angles were calculated by first splitting the N- and C-domain of the transporter into two equal segments representing the intracellular and extracellular halves. α was defined as the angle between the centers of masses of 1) the intracellular half of N-domain, 2) the extracellular half of the protein (including both N- and C-domains), and 3) the intracellular half of C-domain. β is the angle between the centers of masses of 1) the extracellular half of N-domain, 2) the intracellular half of the protein (including both N- and C-domains), and 3) the extracellular half of C-domain. The metadynamics simulation was performed for 200 ns from which a putative O conformation was obtained. The obtained O state is stabilized by the formation of salt bridges at the extracellular and intracellular gates (Supplementary Fig. 14e). The stability of the formed salt bridges and

the intracellular closure were confirmed through a 500-ns equilibrium simulation.

## DEER-guided refinement of the IF and O conformations with reMD simulations

The DEER distance histograms were used to refine the IF and O structures employing restrained-ensemble MD (reMD)[40]. Prior to the simulations, dummy spin labels (OND) were added to the sites of interest using PSFGEN. Pairwise distances between the dummy spin labels were harmonically restrained to the experimental values with a force constant of 10 kcal/mol/$\text{Å}^2$ and then simulated for 1 ns. REMD simulations were performed using the same parameters described in "*Simulation conditions*" except for using a 1-fs timestep for integration.

## Reporting summary

Further information on research design is available in the Nature Research Reporting Summary linked to this article.

## Data availability

Data that support this study are available from the corresponding authors upon reasonable request. The generated data, including those from the DEER experiments, are available in the manuscript or supplementary materials. Source data, including DEER, molecular dynamics trajectories, and DeerA software have been deposited to the Zenodo repository maintained by CERN, https://doi.org/10.5281/zenodo.6678447. Source Data underlying Fig. 8 and Supplementary Figs. 5, 11 and 12 are available as a Source Data file. Source data are provided with this paper.

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

## Acknowledgements

The authors wish to thank Mr. Brant Swiney for his assistance in creating all *Hn*Spns mutants and protein expression and purification. We thank Drs. Joel Eissenberg and Derek P. Claxton for critical reading and editing of the manuscript and Dr. Edwin Antony and Ms. Sahiti Kuppa for assistance with the CD analysis. This work was supported by a Lottie C. Hardy Foundation grant (to R.D.). The molecular dynamics studies were supported by NIH grant P41-104601 (to E.T.), and computing resources were provided by XSEDE grant MCA06N060 (to E.T.) and by Microsoft Azure.

## Author contributions

R.D. and E.T. designed the experiments and simulations. R.D. and S.G. purified the mutants and reconstituted in nanodiscs. R.D. performed the EPR experiments and analyzed the data. A.R. and S.D.G. performed and analyzed the simulations. S.G. performed the thermal stability analyses and cell growth assays. R.D., E.T., A.R., and S.D.G. wrote the paper. A.R and S.D.G. contributed equally to this work.

## Competing interests

The authors declare no competing interests.
