## [Peer Review File · Nature Communications]

Reviewers' Comments:

Reviewer #1:

Remarks to the Author:

The manuscript Proton-driven alternating access in a spinster transporter, an emerging family of broad-specificity efflux pumps by Dastvan and colleagues presents a very interesting study on a previously not well characterized 12-transmembrane helices MSF transporter.

The methods used, namely DEER spectroscopy and DEER data analysis in combination with MD simulations, are very appropriate and the achieved results are clear and highly informative. The study was conducted in systematic way, multiple controls were performed, and the obtained data are highly reliable. The conclusions made are adequate.

This study presents a novel and unique data and interpretation about the functional mechanism of this particular HnSpns transporter, which deviate from the structural models built based on existing high-resolution structure of homologous protein. Indeed, the DEER distances measured between multiple spin-labeled sites on both periplasmic and cytoplasmic side of the protein in detergent solution and lipid environment does provide highly reliable and unique information about functionally-related structures and structural transitions in a membrane protein.

Therefore, although further studies to test the hypothesis and results from this work would be helpful, I believe that the manuscript presented here is of very good quality, provides novel results and would be of great interest to the scientific community working in the field of membrane transporters and broadly membrane proteins. Based on this, I would recommend the publication of this manuscript in Nature Communications.

I have one question/recommendation regarding the missing error bars in the state population vs. pH curves (pKa) (Suppl Figs. 11 and 12) – are the points in the rightmost panes obtained from a single measurement or are average values of multiple measurements? Some evidence about data reproducibility would be helpful.

Reviewer #2:

Remarks to the Author:

The manuscript presents the results of a structural analysis using DEER spectroscopy to explore the dynamics of HnSpns, a bacterial analog of a class of lipid transporters termed Spinster (Spns). The experimental probing of the protein construct with mutated Cys residues to enable probe attachments at various sites was carried out at several pH values (from 4 to 9). The measurements of intramolecular distances in various parts of the membrane protein led to inferences about conformational changes related to protonation states of the system which considered to reflect functionally important states characteristic of MFS transporters. The HnSpns was subjected to MD simulations with the well-tempered metadynamics approach under some of the conditions explored experimentally, in a protocol termed DEER-guided MD simulations that is expected to provide structural refinement from restrained-ensemble simulations.

The ample results from the experimental DEER measurements are well documented in the manuscript and the Suppl Info. Such detailed results are likely to be valued in the field as elements in the continued quest to provide structural and functional information about the human analogs once the extent of structural and functional similarities can be established.

The inference and conclusions from the data presented in the paper are presented in the context of subsections describing the various measurements and results. In these distributed places, such inferences are expressed rather vaguely and without a specific rationale that leads from the finding to the stated conclusion. Some examples are:

-p.5 lines 7-9: Thus, the effect of these protonation-mimetic mutations on the stability of HnSpns could reflect their influence on structural dynamics and functional properties of the transporter.

-p.5 lines 30-32: Overall, these observations underscore the previously unknown role of HnSpns, and due to similar binding pocket properties potentially human Spns proteins, in MDR.
[Note: The attribution of a heretofore unknown physiological role deserves much more elaboration].

-p.6 lines 16-19: Notably, shifts to shorter distances in lipid nanodiscs compared to micelles were observed for distance pairs involving the lipid-interacting helix TM3. Also, the amplitudes of pH-dependent conformational changes were higher in micelles than in nanodiscs.
[Note: It is difficult to interpret the results in Figs. 4.5., and S4 as Support for these assertions, because there seems (I may be missing something) to be good overlap between the distance ranges in the different media, and because it is difficult to perceive what is being inferred from the difference in amplitude for curves centered on nearly identical distance values but differing in ranges (flexibility?).]

-p.6 line 27: Our resistance assays highlight HnSpns, and potentially other Spns proteins, as MDR transporters.
[Note: This assertion deserves some elaboration, perhaps using some elements from the Discussion].

-p.7 lines 32-36: On the periplasmic side, although slight shifts in the conformational equilibria can be observed with different protonation-mimetic mutations, the amplitude of the changes are much smaller than the intracellular side, indicating independent proton-coupled conformational changes on the two sides of the transporter (Fig. 7b and Supplementary Figs. 8, 10).
[Note: It is not clear why the magnitude of the change suggests independence - it just seems to suggest a difference in local interactions, which indeed seems to be the case from the results presented in the subsequent section starting on line 37].

The computational studies are not well described (e.g., was only WT simulated, and then compared to construct C?; which results pertain to what pH values – changes in protonation schemes are not directly interpretable as specific changes in pH values; what were the lengths of various simulations, as this information does not seem to accompany every one, and what were the criteria for such lengths?).

The manuscript also does not make it clear what MD results connect to which specific experimental results, or even more importantly what the computational findings add to its significance. Since the refinement of the structures is based on DEER results, finding that tunnels and rearrangements are compatible with those from the experiments, is not informative. On the other hand, many parts of the manuscript the functional interpretation of the experimental results utilizes information about mutant constructs (e.g., p.9, lines 26-34). Results from MD simulations of the various constructs could probe the relation between in inferences from the behavior of the mutants and the structural observations that are claimed to underlie the mechanisms and add valuable mechanistic understanding that would justify the inclusion of these studies in the manuscript.

Reviewer #3:

Remarks to the Author:

In this manuscript, the authors use a combination of growth assays, DEER studies, and MD simulations to explore the transport mechanism of a bacterial spinster lipid transporter HnSpns. This transporter falls in the MFS family, and one of its primary substrates is sphingosines but it also has a role in multidrug efflux. This family member uses the energy stored in pH gradients to drive transport, and the final model suggests a 2 H⁺:1 ligand stoichiometry, although the possibility of 1:1 is mentioned. H⁺ transport is difficult to study because there are so many potential possibilities for where the protons are moving, making this a difficult cycle to tease apart. You can see the complexity in the final model in Figure 10, which is a little tricky to piece together. This problem is even more difficult given that high resolution structures only exist for the inward-facing conformation. The authors don't have all of the steps nailed down, but the combination of DEER under

basic/neutral/acidic pH values, protonation mimicking mutants, and several MD simulations with different protonation states makes a good attempt at understanding where the protons are going and how they are changing the conformation of the transporter. One interesting result is that the authors suggest that there may not be a standard outward-facing state in this transport cycle. For all of these reason, I think that this paper is an important advance (even if it isn't the final word on HnSpns transport), and it will be appreciated broadly in the protein biophysics community. That said, I have several questions that I would like to be addressed. My two biggest concerns are the reproducibility of the free MD simulations with different protonation states in Figure 6, and the clarity of writing concerning these simulations as well as the overall description of the final transport model. See below for more details.

Primary Comments

No outward-facing state. A key prediction is that this transporter does not adopt an outward-facing state. To this end the R289-R60 salt bridge is predicted to keep the outer gate closed. I really wanted to see mutants at these positions to see if they destabilize the inward facing state or stabilize outward facing, but I was very disappointed to read on Page 7-8 that you can't express R289/D60 mutants. Ugh.

Following up on this and the role of proton entry, what do the MD simulations of your putative occluded state tell you about water access from the extracellular space, the potential for SIP exit to the outer leaflet, or even the ability to accommodate SIP in the occluded state?

Ligand binding and closing the inner gate. Pg 6. Lines 20-27. How do you now that S1P bound? You would have to know this before you state that ligand binding doesn't drive the conformational change. Despite this section with a subtitle that states that ligand binding does not trigger a conformational change, in the caption for your final cartoon model/schematic (Fig. 10) you state:

"Substrate binding following Glu129 protonation triggers closing of the intracellular side and transition to a periplasmic flexible but closed conformation"

Is it Glu129 protonation/deprotonation that drives closing, and sometimes it is captured with the substrate bound? What is biasing the cycle here?

Protonation states of E129 and D41. This section could be more clearly written. When I look at the 4 panels in Figure 6 (a-d) I can clearly see that a-c all show rather stable and small distance values, but comparing panels c to d reveals that protonation of E129 breaks the E129-R42 salt bridge (red), hence suggesting that the major conformational change is controlled by a protonation/deportation event of E129. In the next section, I think you do a more clear job of discussing these results, especially with the sentence:

"However, protonation of Glu129 alone (Fig. 6d) triggers the largest conformational changes in TMs 1 and 4 and enables the proper formation of the charge-relay network."

In Figure 7 - the E129Q mutation shifts the equilibrium towards the pH 4 state closing of the intracellular side is very nice! Do the MD results in Figure 6 for E129 protonated show closing of the inner gate? I don't really see a metric for that. What do the M84-S386 distances do compared to doubly deprotonated in the MD?

Regarding the MD simulations in Figure 6 - I would like to see at least 2 more independent simulations of each of the 4 conditions to make sure that these results are reproducible.

Minor Comments

- Pg 5. Lines 18-21. These two sentences seem to contradict each other (unable in the first and similarly able in the second):

It has been reported that MsSpns is unable to confer resistance against the highly lipophilic drug rifampicin, the most effective, first-line antibiotic against tuberculosis. In our resistance assays, CL and WT HnSpns similarly conferred resistance against different concentrations of the tested compounds, confirming the functionality of the CL construct.

- Pg 5. Lines 52-53. This title is not a statement like the other subtitles. It breaks your style, and I prefer your statement driven titles more than this descriptive subtitle.
Pg 5. Line 53. When you state "Gating TM pairs 2/11 and 5/8 behave differently on the extracellular side (Fig. 4)." I thought you were making a contrast with 2/11 on the intracellular face, but you were contrasting 2/11 and 5/8 both on the extracellular face, right? Perhaps you could be more clear here. You get it in the next sentence, but I was still thinking IF from the last paragraph. This is not a big problem, as this whole section is very nicely written (don't change it too much if you do change it!).
- Pg 6. Lines 44-45. To which mutant does "this mutant" refer?
- Figure 6. Some of the residue labels have white spaces over them.
- In Figure 10b, you should label the key residues in one of the panels.
- Pg 9. You state:
"Based on our DEER distance of the S57-L111 pair (TM1-TM3; Supplementary Fig. 8), we predict these residues act as gatekeepers that sense the proton-dependent structural rearrangements on the extracellular but not on the intracellular side."
You don't really mean that S57 and L111 are "sensing" protons right? They are reporters of what the protons are doing, but not actually sensing.
- Pg 10. "In contrast, Spns2 is predicted to function as an S1P efflux antiporter, given the inwardly directed H⁺ gradient across the cell membrane." This pH gradient is weak (except in cancer cells), right? Something like pH 7.4 to 7.2? If the proton gradient must be high outside compared to the inside, I would expect this bug to live in acidic conditions. Is that true?
- In Figure 9, the dark and light green spheres/tunnels in panels b and c are not described in the figure caption.
- Pg 7. Line 13 "implying population of the IF and O states." – Is this grammatically correct?
- Pg 7. Line 28 "the reverse effect of the D41N mutation" – would you be more explicit by what "reverse effect" means here?
- Pg 8. Line 31 "D41N mutation which mimics Glu129 protonation" How does the D41 protonation mimic also mimic E129 protonation? Is this a misprint?

First Revision:

The supportive and constructive critiques and comments of the reviewers are greatly appreciated and were very helpful in improving the manuscript. The reviewers were unanimous in evaluating the study as of suitable quality and sufficient general interest. Below is a point-by-point response.

Reviewer 1:

I have one question/recommendation regarding the missing error bars in the state population vs. pH curves (pKa) (Suppl Figs. 11 and 12) – are the points in the rightmost panes obtained from a single measurement or are average values of multiple measurements? Some evidence about data reproducibility would be helpful.

Response: We thank the reviewer for noticing the missing error bars. They are added to Figure 8 and Supplementary Figures 11 and 12. The error bars for the state populations represent the uncertainty in the fit parameter reported at the 2σ (95%) confidence level. For titration experiments, due to the sheer number of pH conditions for each mutant, it was impractical to repeat every single sample more than once. However, for all the titration samples, reproducible biological repeats exist for three pH values (pH 4, 7.5 and 9). Overlaid biological repeats are shown for the WT and two background mutations of S25-S386 in lipid nanodiscs.

Reviewer 2:

The inference and conclusions from the data presented in the paper are presented in the context of subsections describing the various measurements and results. In these distributed places, such inferences are expressed rather vaguely and without a specific rationale that leads from the finding to the stated conclusion. Some examples are:

Response: We have revised those conclusions to address the comments of the Reviewer and more clearly articulate the interpretations.

-p.5 lines 7-9: Thus, the effect of these protonation-mimetic mutations on the stability of HnSpns could reflect their influence on structural dynamics and functional properties of the transporter.

Response: The modified sentence is: "Thus, these protonation-mimetic mutations differently influence the stability and potentially structural dynamics of HnSpns."

-p.5 lines 30-32: Overall, these observations underscore the previously unknown role of HnSpns, and due to similar binding pocket properties potentially human Spns proteins, in MDR. [Note: The attribution of a heretofore unknown physiological role deserves much more elaboration].

Response: We agree with the reviewer regarding more elaboration. The brevity to some extent was a consequence of the available space. The modified sentence is: "The mechanistic significance of

these observations is further elaborated in Discussion. Overall, these observations underscore the previously unknown role of *HnSpns*, and due to similar binding pocket properties potentially human *Spns* proteins, as broad-specificity efflux pumps that could mediate clinically relevant MDR to antibiotics and chemotherapeutic drugs. Many MDR transporters from MATE family (e.g., pfMATE) and ABC superfamily (e.g., MsbA, ABCB1, ABCC1, and ABCG2) also physiologically function as lipid flippases/transporters. Interestingly, ABCC1 and ABCG2 also participate in S1P transport.⁴⁴

-p.6 lines 16-19: Notably, shifts to shorter distances in lipid nanodiscs compared to micelles were observed for distance pairs involving the lipid-interacting helix TM3. Also, the amplitudes of pH-dependent conformational changes were higher in micelles than in nanodiscs. [Note: It is difficult to interpret the results in Figs. 4.5., and S4 as Support for these assertions, because there seems (I may be missing something) to be good overlap between the distance ranges in the different media, and because it is difficult to perceive what is being inferred from the difference in amplitude for curves centered on nearly identical distance values but differing in ranges (flexibility?).]

Response: This statement regarding the difference between some DEER results in LMNG micelles and lipid nanodiscs is unnecessary and unrelated to the rest of the section and has been removed.

*-p.6 line 27: Our resistance assays highlight *HnSpns*, and potentially other *Spns* proteins, as MDR transporters. [Note: This assertion deserves some elaboration, perhaps using some elements from the Discussion].*

Response: The revised sentence is: "Our resistance assays highlight *HnSpns*, and potentially human *Spns* proteins, as broad-specificity efflux pumps with the possibility of the latter being involved in cancer MDR⁸ (see Discussion)."

-p.7 lines 32-36: On the periplasmic side, although slight shifts in the conformational equilibria can be observed with different protonation-mimetic mutations, the amplitude of the changes are much smaller than the intracellular side, indicating independent proton-coupled conformational changes on the two sides of the transporter (Fig. 7b and Supplementary Figs. 8, 10). [Note: It is not clear why the magnitude of the change suggests independence - it just seems to suggest a difference in local interactions, which indeed seems to be the case from the results presented in the subsequent section starting on line 37].

Response: The sentence is revised to: "On the periplasmic side, only slight and negligible shifts in the conformational equilibria can be observed in response to different protonation-mimetic mutations, indicating these proton switches mainly control conformational changes on the intracellular side of the transporter (Fig. 7b and Supplementary Figs. 8, 10)."

The computational studies are not well described (e.g., was only WT simulated, and then compared to construct CL?;

Response: The WT protein was employed in our simulations, and four independent replicates per each of the four protonation states. These are now clarified in the Methods section.

which results pertain to what pH values – changes in protonation schemes are not directly interpretable as specific changes in pH values;

Response: To better articulate the correspondence of the four MD simulations using different protonation states of protonation switches and experimental DEER data using their protonation-mimetic mutations at different pH values, the results section is expanded:

p.7, lines 35-43: "Our DEER data on protonation-mimetic mutations (four different permutations) at basic pH values (e.g., pH 9.0) support the results of the MD simulations (Fig. 7 and Supplementary

Figs. 7 and 8). Under basic pH conditions, proton switches other than the mutated residues are expected to be mainly deprotonated, facilitating the correspondence of the DEER data (Fig. 7) and MD simulations (Fig. 6). However, equilibrium MD simulations (Fig. 6) probe the local protonation-dependent charge interactions and conformational changes (e.g., formation of the NTD charge-relay network [lower panels], rotation/bending of TM helices [middle panels]). The ensuing global changes (e.g., closing of the intracellular side) are probed using DEER distance pairs between NTD and CTD (e.g., M84-S386, S25-S386; Fig. 7a)."

what were the lengths of various simulations, as this information does not seem to accompany every one, and what were the criteria for such lengths?).

Response: The length of MD simulations is now added (p.6, line 47) and were stated in the Methods section and shown in Figure 6. As stated above, the 500 ns simulations in lipid membranes of four different protonation permutations (each independently repeated four times: $4 \times 4 \times 500 \text{ ns} = 8 \mu\text{s}$) are adequate to probe local proton-dependent charge interactions or conformational changes, triggering the global transitions.

The manuscript also does not make it clear what MD results connect to which specific experimental results, or even more importantly what the computational findings add to its significance. Since the refinement of the structures is based on DEER results, finding that tunnels and rearrangements are compatible with those from the experiments, is not informative. On the other hand, many parts of the manuscript the functional interpretation of the experimental results utilizes information about mutant constructs (e.g., p.9, lines 26-34). Results from MD simulations of the various constructs could probe the relation between in inferences from the behavior of the mutants and the structural observations that are claimed to underlie the mechanisms and add valuable mechanistic understanding that would justify the inclusion of these studies in the manuscript.

Response: We agree with the reviewer on the valuable mechanistic understandings inferred from MD simulations which were performed to provide atomic-scale frameworks for the observed experimental distances. We have now revised corresponding sections to better articulate the relation to experimental data as summarized below:

1- Proposed energy coupling mechanism is mainly deduced from these MD simulations and supported by DEER data on the protonation-mimetic mutations:

p. 7, lines 8-12: "As a plausible coupling mechanism, protonation of Glu129 is promoted by decreasing the nearby positive electrostatic potential, either by compensating the positive charge of the conserved Arg122, e.g., by formation of a charge-charge interaction with deprotonated Asp41, or by binding of the negatively charged substrate to Arg42." and elaborated in that section.

p. 9, lines 39-44: "Using an integrated computational and spectroscopic approach, the extensive results presented here reveal new mechanistic details of *HnSpns* as a close homolog of the human *Spns* proteins, including its conformational dynamics following protonation/deprotonation of Glu129 which is regulated by the protonation state of Asp41 and substrate binding. Our comprehensive MD simulations implicate neutralization of the highly conserved Arg122 or Arg42 in this regulatory mechanism."

p. 10, lines 42-44: "This model identifies residues critical for protonation and its regulation, reveals how sequential protonation of these switches drives the conformational transitions, and suggests a mechanism of energy coupling to substrate transport".

2- Local proton-dependent charge interactions and conformational changes triggering the global transitions (e.g., closing of the intracellular side) are also directly deduced from MD simulations (Fig. 6, middle and lower panels) and supported by DEER data (Fig. 7). They are detailed in the results section titled “Proton-dependent changes occur preserving the extracellular-closed state” (p. 7, line 18) and compared to DEER data in the following section (p. 7, line 35).

3- Modeling the novel occluded state and the key stabilizing interactions. The central role of the conserved periplasmic salt bridge (Asp60^{TM2}:Arg289^{TM7}) in stabilizing the occluded state was directly deduced from different MD simulations and supported by DEER results on distance pairs between TM7 and TM1/TM2 or other helices like TM3 (Fig. 5, distance pairs S57-M288 and L111-M288).

Reviewer 3:

That said, I have several questions that I would like to be addressed. My two biggest concerns are the reproducibility of the free MD simulations with different protonation states in Figure 6, and the clarity of writing concerning these simulations as well as the overall description of the final transport model.

Response: Regarding the reproducibility of the MD simulations, each of the 500-ns simulations of the four different protonation permutations in lipid membranes were independently repeated three times and represented in Supplementary Figure 6. We added an additional repeat to each system, now reporting four independent copies for each of the different protonation permutations (4 replicates × 4 protonation permutations × 500 ns). On the protonation-dependent charge interactions (i.e., Glu129^{TM4}:Arg42^{TM1} and Asp41^{TM1}:Arg122^{TM4}) at the center of the proposed coupling mechanism and transport model, the MD results are reproducible. Some variations can be observed which stem from the stochastic nature of these interactions. A comment on this is now added to the section (p. 7, lines 12-15).

No outward-facing state. A key prediction is that this transporter does not adopt an outward-facing state. To this end the R289-R60 salt bridge is predicted to keep the outer gate closed. I really wanted to see mutants at these positions to see if they destabilize the inward facing state or stabilize outward

facing, but I was very disappointed to read on Page 7-8 that you can't express R289/D60 mutants. Ugh.

Response: The disappointment is mutual! We tested different lipid compositions using S57-M288 distance pair that monitors the extracellular opening between TM1/2 and TM7. We did not observe an extracellular opening and no significant lipid-induced shifts in equilibria were observed.

Following up on this and the role of proton entry, what do the MD simulations of your putative occluded state tell you about water access from the extracellular space, the potential for SIP exit to the outer leaflet, or even the ability to accommodate SIP in the occluded state?

Response: We calculated the water density for the last 250 ns of the occluded state equilibrium trajectory. The 2D histogram is shown below. Regarding the proton entry and access to the main intracellular proton switches in the NTD, as apparent, a pathway from the periplasmic side to Asp41 and Glu129 via the main vestibule rather than through the putative tunnel is possible. Also, a proton

transfer pathway is possible from the main vestibule to the acidic residues in the NTD charge-relay

network (e.g., Asp142) that stabilizes the intracellular-closed conformation. The main vestibule in the occluded state could accommodate the substrate. These remarks are added to the manuscript (p. 9, line 7-12) and this figure is added to the Supplementary Fig. 14. The intrinsic flexibility of the protonated periplasmic side deduced from DEER experiments at pH 4 (lower panel) allows for S1P exit through the putative periplasmic gate between TMs 5/8. The putative periplasmic path for proton translocation is constricted in the occluded model. Though, based on our DEER distance of the S57-L111 pair as reporter (TM1-TM3; Supplementary Fig. 8) as well as conservation analysis of the residues along this path, it could provide an alternative pathway for proton entry.

Ligand binding and closing the inner gate. Pg 6. Lines 20-27. How do you now that S1P bound? You would have to know this before you state that ligand binding doesn't drive the conformational change.

Response: S1P binding to *HnSpns* is not examined. Though, using the well-studied distance pair S25-S386 (TM1-TM10/11) on the intracellular side, no significant substrate-induced shift in equilibria was observed for capreomycin and ethidium bromide as known *HnSpns* substrates with the ability to confer resistance against them. This section is extensively rewritten (p. 6, lines 28-37) and extensive supporting experimental data are added to the revised Supplementary Fig. 5.

Despite this section with a subtitle that states that ligand binding does not trigger a conformational change, in the caption for your final cartoon model/schematic (Fig. 10) you state: "Substrate binding following Glu129 protonation triggers closing of the intracellular side and transition to a periplasmic flexible but closed conformation". Is it Glu129 protonation/deprotonation that drives closing, and sometimes it is captured with the substrate bound? What is biasing the cycle here?

Response: This is a valid point. Though, protonation of Glu129 can be promoted/sustained by decreasing the nearby positive electrostatic potential through binding of the negatively charged substrate to Arg42. The statement is revised to: "Proton transfer from Asp41 to Glu129 liberates Arg42 to bind the substrate, e.g., S1P **(3)**. Glu129 protonation triggers closing of the intracellular side and transition to a periplasmic flexible but closed conformation **(4)**."

Protonation states of E129 and D41. This section could be more clearly written. When I look at the 4 panels in Figure 6 (a-d) I can clearly see that a-c all show rather stable and small distance values but comparing panels c to d reveals that protonation of E129 breaks the E129-R42 salt bridge (red), hence suggesting that the major conformational change is controlled by a protonation/deportation event of E129. In the next section, I think you do a more clear job of discussing these results, especially with the sentence:

"However, protonation of Glu129 alone (Fig. 6d) triggers the largest conformational changes in TMs 1 and 4 and enables the proper formation of the charge-relay network."

Response: The emphasis and a novel aspect of this section is the regulatory mechanism of Glu129 protonation that was deduced from the MD simulations. We modified the title to "Protonation of Asp41 regulates the intracellular master switch Glu129", to emphasize the central role of Glu129 protonation in the conformational transition. An energy coupling mechanism emerges from these studies and summarized as (p. 7, lines 8-12): "As a plausible coupling mechanism, protonation of Glu129 is promoted by decreasing the nearby positive electrostatic potential, either by compensating the positive charge of the conserved Arg122, e.g., by formation of a charge-charge interaction with deprotonated Asp41, or by binding of the negatively charged substrate to Arg42."

In Figure 7 - the E129Q mutation shifts the equilibrium towards the pH 4 state closing of the intracellular side is very nice! Do the MD results in Figure 6 for E129 protonated show closing of the inner gate? I don't really see a metric for that. What do the M84-S386 distances do compared to doubly deprotonated in the MD?

Response: The equilibrium MD simulations are suitable to probe local proton-dependent charge interactions or conformational changes that trigger the global transitions. Closing of the intracellular side was only captured using metadynamics simulations and probed by DEER distance pairs between NTD and CTD (M84-S386, S25-S386). The DEER on M84-S386 at pH 9 indicated that both intracellular-closed and open populations are sampled, and the equilibrium shifts to the intracellular-closed state at acidic pH values. The MD simulations on the doubly deprotonated state revealed that under this condition the charge-relay network between Asp142^{TM4}-Arg91^{TM3}-Asp87^{TM2} in the NTD could form and therefore sampling the intracellular-closed conformation is still possible. Accordingly, in two of the four independent simulation copies of the doubly deprotonated state (Supplementary Figure 6c), we observe sampling of the dissociated Glu129:Arg42 interaction. Though, in all four copies Asp41:Arg122 salt-bridge remains intact. The manuscript is revised (p. 7, lines 44-51).

Regarding the MD simulations in Figure 6 - I would like to see at least 2 more independent simulations of each of the 4 conditions to make sure that these results are reproducible.

Response: Regarding the reproducibility of the MD simulations, each of the 500 ns simulations of the four different protonation permutations in lipid membranes were independently repeated three times and represented in Supplementary Figure 6. We added an additional repeat to now have four independent copies of each of the different protonation permutations (4 replicates × 4 protonation permutations × 500 ns). This is now more clearly stated in the revised manuscript. On the protonation-dependent charge interactions (i.e., Glu129^{TM4}:Arg42^{TM1} and Asp41^{TM1}:Arg122^{TM4}) at the center of the proposed coupling mechanism and transport model, these MD results are reproducible. Some variations can be observed which stem from the stochastic nature of these interactions.

• Pg 5. Lines 18-21. These two sentences seem to contradict each other (unable in the first and similarly able in the second):

It has been reported that MsSpns is unable to confer resistance against the highly lipophilic drug rifampicin, the most effective, first-line antibiotic against tuberculosis. In our resistance assays, CL and WT HnSpns similarly conferred resistance against different concentrations of the tested compounds, confirming the functionality of the CL construct.

Response: The first sentence is about the *Mycobacterium smegmatis* MDR efflux transporter MSMEG_3705 (*MsSpns*) and the following sentence is about similar behavior of the WT and CL construct of *HnSpns*. The issue is resolved by separating the sentences.

• Pg 5. Lines 52-53. This title is not a statement like the other subtitles. It breaks your style, and I prefer your statement driven titles more than this descriptive subtitle.

Response: Thanks. The title is revised to “Protonation induces a flexible outward-closed conformation”.

Pg 5. Line 53. When you state “Gating TM pairs 2/11 and 5/8 behave differently on the extracellular side (Fig. 4).” I thought you were making a contrast with 2/11 on the intracellular face, but you were contrasting 2/11 and 5/8 both on the extracellular face, right? Perhaps you could be more clear here. You get it in the next sentence, but I was still thinking IF from the last paragraph. This is not a big problem, as this whole section is very nicely written (don’t change it too much if you do change it!).

Response: The modified sentence is: “On the extracellular side, gating TM pairs 2/11 and 5/8 behave differently under acidic conditions (Fig. 4).”

• Pg 6. Lines 44-45. To which mutant does “this mutant” refer?

Response: The sentence is revised to: “This might explain the higher thermal stability of the D41N/E129Q mutant (Supplementary Figs. 1c and 1d).”

• Figure 6. Some of the residue labels have white spaces over them.

Response: The figure is modified.

• In Figure 10b, you should label the key residues in one of the panels.

Response: The figure is modified.

• Pg 9. You state: “Based on our DEER distance of the S57-L111 pair (TM1-TM3; Supplementary Fig. 8), we predict these residues act as gatekeepers that sense the proton-dependent structural rearrangements on the extracellular but not on the intracellular side.” You don’t really mean that S57 and L111 are “sensing” protons right? They are reporters of what the protons are doing, but not actually sensing.

Response: The reviewer is correct. We have modified the sentence to: “Based on our DEER distance of the S57-L111 pair as reporter (TM1-TM3; Supplementary Fig. 8), we predict these residues act as gatekeepers ...”

• Pg 10. “In contrast, Spns2 is predicted to function as an S1P efflux antiporter, given the inwardly directed H⁺ gradient across the cell membrane.” This pH gradient is weak (except in cancer cells), right? Something like pH 7.4 to 7.2? If the proton gradient must be high outside compared to the inside, I would expect this bug to live in acidic conditions. Is that true?

Response: As reviewer mentioned, extracellular acidosis that is typically found in inflammatory tissues or tumors might stimulate the Spns2 activity and enhance S1P export. Hence, the pH gradient might be a key regulatory factor. *HnSpns* is a gram negative marine prosthecate.

• In Figure 9, the dark and light green spheres/tunnels in panels b and c are not described in the figure caption.

Response: The figure caption is modified.

• Pg 7. Line 13 “implying population of the IF and O states.” – Is this grammatically correct?

Response: The sentence is revised to: “...implying presence of the IF and O states.”

• Pg 7. Line 28 “the reverse effect of the D41N mutation” – would you be more explicit by what “reverse effect” means here?

Response: The modified sentence is: “..., the reverse effect of the D41N mutation compared with the E129Q, can be observed in both DEER and CW EPR data”.

• Pg 8. Line 31 “D41N mutation which mimics Glu129 protonation” How does the D41 protonation mimic also mimic E129 protonation? Is this a misprint?

Response: The modified sentence is: “...the predicted pK_a for Glu129 in the O state matches the experimental intracellular pK of conformational changes in lipid nanodiscs for the D41N mutation which Asp41 is removed as a protonation site and the pK_a therefore is related to the Glu129 protonation reaction (Fig. 8c).”

Reviewers' Comments:

Reviewer #1:

Remarks to the Author:

The Authors addressed my comments positively. Therefore, I recommend the manuscript for publication.

Reviewer #2:

Remarks to the Author:

The main conceptual/critical issues raised in the previous review have been addressed in the rebuttal and the manuscript. This includes the need to detail and explain more fully the contribution of the MD simulations which was flagged by more than one Reviewer.

Unfortunately, the replacement sentences for some of the statements that were confusing in the original manuscript, have not added clarity. This is the case for p.5 lines 7-9 which was replaced by: The modified sentence is: "Thus, these protonation-mimetic mutations differently influence the stability and potentially structural dynamics of HnSpns." which is still unclear and confusing (is it that different protonation sites trigger different combinations of effects? but this would be trivial)
p.5 lines 30-32: The revision states "Overall, these observations underscore the previously unknown role of HnSpns," which does not make any sense because the results cannot emphasize the absence of some knowledge about physiological function.
This is a minor concern and may be mitigated by reconsidering these statements and/or eliminate them altogether.

In summary, results for this system remain important, but the manuscript remains unclear, convoluted, and does not present the work and the conclusions in the best scholarly manner. If some attention could be paid to this, it would greatly enhance the value of the work for the community and improve its scientific impact.

Reviewer #3:

Remarks to the Author:

The authors have addressed all of my concerns. I had missed Supplemental Figure 6, and did not appreciate that 3 simulations had been run already. I am happy to see an additional one was performed.

Second Revision:

We are gratified that our response was well received and clarified the previous concerns. In response to reviewer 2's comments, we are submitting a revised version that addresses the reviewer's concerns and follows the guidelines.

Below is a point-by-point response. Corresponding changes have been introduced in the revised manuscript and highlighted in yellow.

Reviewer 2:

Unfortunately, the replacement sentences for some of the statements that were confusing in the original manuscript, have not added clarity. This is the case for p.5 lines 7-9 which was replaced by: The modified sentence is: "Thus, these protonation-mimetic mutations differently influence the stability and potentially structural dynamics of HnSpns." which is still unclear and confusing (is it that different protonation sites trigger different combinations of effects? but this would be trivial).

Response: This statement refers to the influence of protonation-mimetic mutations on thermal stability of *HnSpns*. For instance, the double mutation of D41N and E129Q compared with E129Q mutation alone increases the T_m in both LMNG micelles and lipid nanodiscs (Supplementary Figs. 1c and 1d and Supplementary Table 1). This increased thermal stability correlates with our MD simulations and DEER experiments of this double mutation. In our MD simulations, double

protonation of these acidic residues stabilizes a distinct conformation (Fig. 6b and Supplementary Fig. 6b) in which the side chain of Arg122 continuously interacts with the backbone oxygen of Arg42 but only transiently with the backbone oxygen of Asp41 (Supplementary Fig. 6e), thereby restricting the rotational degree of freedom of TM1, which is essential for both substrate binding and the global structural transition. This might explain the higher thermal stability of the D41N/E129Q mutant. Also, in our DEER experiments of the S25-S386 (TM1-TM10/11) intracellular pair, the D41N-E129Q double mutation completely abrogates the proton-dependent conformational changes on the intracellular side.

The statement is revised to: "Among protonation-mimetic mutations, D41N/E129Q and E129Q mutation alone have the highest and the lowest T_m values in either micelles or lipid nanodiscs, respectively (Supplementary Figs. 1c and 1d and Supplementary Table 1). So, introducing the D41N mutation combined with E129Q enhances the thermal stability of *HnSpns*. D142N mutation has a similar effect in nanodiscs. D142N putatively stabilizes the IF conformation by interrupting the structural motif A-associated charge-relay network²⁰. Thus, these protonation-mimetic mutations differentially influence the thermal stability and potentially structural dynamics of *HnSpns*."

p.5 lines 30-32: The revision states "Overall, these observations underscore the previously unknown role of HnSpns," which does not make any sense because the results cannot emphasize the absence of some knowledge about physiological function. This is a minor concern and may be mitigated by reconsidering these statements and/or eliminate them altogether.

Response: The reviewer has a fair point, that the results don't "underscore" or emphasize the absence of knowledge, but rather they "disclose," "reveal" or "argue for" a role for these proteins that was previously unknown.

The sentence in the abstract is revised to: "Furthermore, our resistance assays reveal substrate polyspecificity and *HnSpns* multidrug resistance (MDR) activity that argue for a previously unknown role of Spns proteins in MDR, beyond their activity in sphingolipid transport and signaling."

The sentence in the results section (p.5, line 32) is revised to: "Overall, these observations disclose a previously unknown role of HnSpns, and due to similar binding pocket properties potentially human Spns proteins, as broad-specificity efflux pumps that could mediate clinically relevant MDR to antibiotics and chemotherapeutic drugs."